# Gradient-Free Analytical Fisher Information of Diffused Distributions

## Abstract

Diffusion models (DMs) have demonstrated powerful distributional modeling capabilities by matching the first-order score of diffused distributions. Recent advancements have explored incorporating the second-order Fisher information, defined as the negative Hessian of log-density, into various downstream tasks and theoretical analysis of DMs. However, current practices often overlook the inherent structure of diffused distributions, accessing Fisher information via applying auto-differentiation to the learned score network. This approach, while straightforward, leaves theoretical properties unexplored and is time-consuming. In this paper, we derive the analytical formulation of Fisher information (AFI) by applying consecutive differentials to the diffused distributions. As a result, AFI takes a gradient-free form of a weighted sum (or integral) of outer-products of the score and initial data. Based on this formulation, we propose two algorithmic variants of AFI for distinct scenarios. When evaluating the AFI's trace, we introduce a parameterized network to learn the trace. When AFI is applied as a linear operator, we present a training-free method that simplifies it into several inner-product calculations. Furthermore, we provide theoretical guarantees for both algorithms regarding convergence analysis and approximation error bounds. Additionally, we leverage AFI to establish the first general theorem for the optimal transport property of the diffusion ODE deduced map. Experiments in likelihood evaluation and adjoint optimization demonstrate the superior accuracy and reduced time-cost of the proposed algorithms.

## 1 Introduction

The emerging diffusion models (DMs) Sohl-Dickstein et al. (2015); Ho et al. (2020); Song & Ermon (2019); Song et al. (2020), generating samples of data distribution from initial noise by learning a reverse diffusion process, have been proven to be an effective technique for modeling data distribution, especially in generating high-quality images Nichol et al. (2022); Dhariwal & Nichol (2021a); Saharia et al. (2022); Ramesh et al. (2022); Rombach et al. (2022); Ho et al. (2022). The training process of DMs can be seen as employing a neural network to match the first-order score $\nabla_{\boldsymbol{x}} \log q_t(\boldsymbol{x})$ of the diffused distributions at varying noise levels.

Recently, there has been a growing trend to recognize the importance of the Fisher information in DMs, defined as the negative Hessian of the diffused distributions' log-density, $-\nabla_{\boldsymbol{x}}^2 \log q_t(\boldsymbol{x})$. The Fisher information provides valuable second-order information of DMs and plays a crucial role in likelihood evaluation (Lu et al., 2022a; Zheng et al., 2023), adjoint optimization (Pan et al., 2023a;b; Blasingame & Liu, 2024), and optimal transport analysis Zhang et al. (2024a).

However, current practices (Sanchez et al., 2022; Song & Lai, 2024) typically overlook the inherent structure of diffused distributions, and access the Fisher information by applying auto-differentiation to the score network. While this is a straightforward approach, it leads to time-consuming gradient operations, even with the help of the Jacobian-vector-product (JVP) technique. Moreover, in the likelihood evaluation task, the current JVP method still needs a quadratic order time complexity concerning the dimension to calculate the trace of the Fisher information, rendering likelihood evaluation intractable for SD-level models. Additionally, due to a lack of comprehensive understanding of Fisher Information, Zhang et al. (2024a) have to impose stringent assumptions to characterize the optimal transport property of the diffusion ODE deduced map.

In this paper, we delve deeper into the inherent quadratic structure of diffused distributions and derive the analytical form of Fisher information (AFI) by applying the consecutive partial differential chain rule to the marginal distributions. Notice that, while this inherent structure has been consistently utilized in the learning process of the score network, it has often been overlooked when accessing Fisher Information up until now. The main contributions of our paper are listed as follows:

- We develop the first analytical formulation of the Fisher Information (AFI) of diffused distributions, which is gradient-free. Initially, we show that the AFI manifests as a weighted summation of outer-products of the score and initial data when the initial distribution is a sum of Dirac. We then extend this result to an integral form in a more general setting. The AFI suggests a theoretical possibility of accessing Fisher information without needing costly gradient calculations in practice.
- Based on the AFI, we propose two algorithmic alternatives to the JVP, each tailored to different types of Fisher Information access. For the evaluation of Fisher information's trace, we introduce a parameterized network to learn the trace, significantly reducing the time complexity of trace evaluation from quadratic to linear w.r.t. the dimension. In scenarios where Fisher Information is applied as a linear operator, we present a training-free method that simplifies the complex linear transformation calculations into several simple inner-product calculations. Furthermore, we provide theoretical guarantees for these algorithms, including convergence analysis and approximation error bounds.
- Utilizing the analytical knowledge of the Fisher information, we establish the first theorem that allows the general diffusion ODE deduced mapping to possess the optimal transport property, eliminating the need for stringent assumptions.

We evaluate our AFI algorithms on likelihood evaluation and adjoint optimization tasks. The empirical results demonstrate the enhanced accuracy and reduced time-cost of our AFI methods.

## 2 PRELIMINARIES

**Notation**. The Euclidean norm over $\mathbb{R}^d$ is denoted by $\|\cdot\|$, and the Euclidean inner product is denoted by $\langle\cdot|\cdot\rangle$. Throughout, we simply write $\int g$ to denote the integral with respect to the Lebesgue measure: $\int g(x)\mathrm{d}x$. When the integral is with respect to a different measure $\mu$, we explicitly write $\int g\mathrm{d}\mu$. When clear from context, we sometimes abuse notation by identifying a measure $\mu$ with its Lebesgue density. We also use $\delta(\cdot)$ to denote the Dirac Delta function.

### 2.1 DIFFUSION MODELS AND DIFFUSION SDES

Suppose that we have a d-dimensional random variable $\boldsymbol{x}_0 \in \mathbb{R}^d$ following an unknown target distribution $q_0(\boldsymbol{x}_0)$. Diffusion Models (DMs) define a forward process $\{\boldsymbol{x}_t\}_{t\in[0,T]}$ with $T > 0$ starting with $\boldsymbol{x}_0$, such that the distribution of $\boldsymbol{x}_t$ conditioned on $\boldsymbol{x}_0$ satisfies

$$\text{(Diffusion Transition Kernel)} \qquad q_{t|0}(\boldsymbol{x}_t|\boldsymbol{x}_0) = \mathcal{N}(\boldsymbol{x}_t; \alpha(t)\boldsymbol{x}_0, \sigma^2(t)\mathbf{I}), \qquad (1)$$

where $\alpha(\cdot), \sigma(\cdot) \in \mathcal{C}([0,T], \mathbb{R}^+)$ have bounded derivatives, and we denote them as $\alpha_t$ and $\sigma_t$ for simplicity. The choice for $\alpha_t$ and $\sigma_t$ is referred to as the noise schedule of a DM. According to Kingma et al. (2021); Karras et al. (2022), with some assumption on $\alpha(\cdot)$ and $\sigma(\cdot)$, the forward process can be modeled as a linear SDE which is also called Ornstein–Uhlenbeck process:

$$\mathrm{d}\boldsymbol{x}_t = f(t)\boldsymbol{x}_t\mathrm{d}t + g(t)\mathrm{d}B_t, \qquad (2)$$

where $B_t$ is the standard d-dimensional Brownian Motion (BM), $f(t) = \frac{\mathrm{d}\log\alpha_t}{\mathrm{d}t}$ and $g^2(t) = \frac{\mathrm{d}\sigma_t^2}{\mathrm{d}t} - 2\frac{\mathrm{d}\log\alpha_t}{\mathrm{d}t}\sigma_t^2$. Under some regularity conditions, the above forward SDE equation 2 have a reverse SDE from time $T$ to 0, which starts from $\boldsymbol{x}_t$ Anderson (1982):

$$\mathrm{d}\boldsymbol{x}_t = \left[f(t)\boldsymbol{x}_t - g^2(t)\nabla_{\boldsymbol{x}_t}\log q(\boldsymbol{x}_t, t)\right]\mathrm{d}t + g(t)\mathrm{d}\tilde{B}_t, \qquad (3)$$

where $\tilde{B}_t$ is the reverse-time Brownian motion and $q(\boldsymbol{x}_t, t)$ is the single-time marginal distribution of the forward process. In practice, DMs Ho et al. (2020); Song et al. (2020) use $\boldsymbol{\varepsilon}_\theta(\boldsymbol{x}_t, t)$ to estimate $-\sigma(t)\nabla_{\boldsymbol{x}_t}\log q(\boldsymbol{x}_t, t)$ and the parameter $\theta$ is optimized by the following objective:

$$\theta^* = \arg\min_\theta \mathbb{E}_t\left\{\lambda_t\mathbb{E}_{x_0, x_t}\left[\|s_\theta(x_t, t) - \nabla_{x_t}\log p(x_t, t|x_0, 0)\|^2\right]\right\}, \qquad (4)$$

where $s_\theta$ represents the parameterized score, i.e., $s_\theta(\boldsymbol{x}_t, t) = -\frac{\boldsymbol{\varepsilon}_\theta(\boldsymbol{x}_t, t)}{\sigma_t}$. This familiar parameterization is called $\epsilon$-prediction. There are also $\boldsymbol{y}$-prediction and $\boldsymbol{v}$-prediction Salimans & Ho (2022). The corresponding loss is equal to replace the term $|\epsilon - \boldsymbol{\varepsilon}_\theta(\boldsymbol{x}_t, t)|$ with $\frac{\alpha_t}{\sigma_t}|\boldsymbol{x}_0 - \bar{\boldsymbol{y}}_\theta(\boldsymbol{x}_t, t)|$ and $|\alpha_t\epsilon - \sigma_t\boldsymbol{x}_0 - \boldsymbol{v}_\theta(\boldsymbol{x}_t, t)|$. The learned $\boldsymbol{\varepsilon}_\theta(\boldsymbol{x}_t, t)$ can be also transformed to a $\boldsymbol{y}$-prediction form by $\bar{\boldsymbol{y}}_\theta(\boldsymbol{x}_t, t) = \frac{\boldsymbol{x}_t - \sigma_t\boldsymbol{\varepsilon}_\theta(\boldsymbol{x}_t, t)}{\alpha_t}$.

## 2.2 Diffusion Models Inference as Neural ODE

It is noted that the reverse diffusion SDE in equation 3 has an associated probability flow ODE (also called diffusion ODE), which is a deterministic process that shares the same single-time marginal distribution Song et al. (2020):

$$\text{(PF-ODE)} \qquad \mathrm{d}\boldsymbol{x}_t = \left[ f(t)\boldsymbol{x}_t - \frac{1}{2}g^2(t)\nabla_{\boldsymbol{x}_t} \log q_t(\boldsymbol{x}_t, t) \right] \mathrm{d}t. \tag{5}$$

By replacing the score function in equation 5 with the noise predictor $\boldsymbol{\varepsilon}_\theta$, the inference process of DMs can be constructed by the following neural ODE:

$$\frac{\mathrm{d}\boldsymbol{x}_t}{\mathrm{d}t} = \boldsymbol{h}_\theta(\boldsymbol{x}_t, t) := f(t)\boldsymbol{x}_t + \frac{g^2(t)}{2\sigma_t}\boldsymbol{\epsilon}_\theta(\boldsymbol{x}_t, t), \quad \boldsymbol{x}_T \sim \mathcal{N}\left(\boldsymbol{0}, \sigma_T^2\boldsymbol{I}\right) \tag{6}$$

## 2.3 Fisher Information in Diffusion Models

The Fisher information matrix in DMs is defined as the negative Hessian of the marginal log-density function, which takes the following matrix-valued form Song et al. (2021); Song & Lai (2024):

$$\boldsymbol{F}_t(\boldsymbol{x}_t, t) := -\frac{\partial^2}{\partial \boldsymbol{x}_t^2} \log q_t(\boldsymbol{x}_t, t) \tag{7}$$

The current technique typically approximately accesses to the Fisher information by accessing the scaled Jacobian matrix of the learned score estimator network $\boldsymbol{\varepsilon}_\theta$:

$$\begin{aligned}
\boldsymbol{F}_t(\boldsymbol{x}_t, t) &= -\frac{\partial}{\partial \boldsymbol{x}_t}\left(\frac{\partial}{\partial \boldsymbol{x}_t} \log p(x_t, t)\right) \\
&\approx -\frac{\partial}{\partial \boldsymbol{x}_t}\left(-\frac{\boldsymbol{\varepsilon}_\theta(\boldsymbol{x}_t, t)}{\sigma_t}\right) = \frac{1}{\sigma_t}\frac{\partial \boldsymbol{\varepsilon}_\theta(\boldsymbol{x}_t, t)}{\partial \boldsymbol{x}_t}
\end{aligned} \tag{8}$$

The full Fisher information matrix within DMs cannot be obtained due to dimensional constraints. For instance, the Stable Diffusion-1.5 model (Rombach et al., 2022) features a latent dimension of $d = 4 \times 64 \times 64 = 16384$, resulting in a Fisher matrix of $16384 \times 16384$. Fortunately, for applications that only need to access the trace or multiplication of Fisher information, it is feasible to use Jacobian-vector-product (JVP) to access Fisher information. For any $d$-dimensional vector $\boldsymbol{v}$, the approximation of $\boldsymbol{v}$ left multiplied by $\boldsymbol{F}_t(\boldsymbol{x}_t, t)$ using JVP is as follows:

$$\text{(JVP)} \qquad \boldsymbol{F}_t(\boldsymbol{x}_t, t)\boldsymbol{v} \approx \frac{1}{\sigma_t}\frac{\partial \boldsymbol{\varepsilon}_\theta(\boldsymbol{x}_t, t)}{\partial \boldsymbol{x}_t}\boldsymbol{v} = \frac{1}{\sigma_t}\frac{\partial\left[\langle \boldsymbol{\varepsilon}_\theta(\boldsymbol{x}_t, t)|\boldsymbol{v}\rangle\right]}{\partial \boldsymbol{x}_t} \tag{9}$$

The JVP is a time-consuming process due to its requirement for gradient calculations within the neural network. In addition, empirical evidence from synthetic distributions, as demonstrated in Lu et al. (2022a), shows that the approximation results from the JVP significantly deviate from the true underlying Fisher information. To our knowledge, there is no theoretical guarantee that Fisher information can be accurately accessed through the JVP. Moreover, the JVP fails to provide any theoretical insight into the Fisher information of diffused distributions.

## 3 Analytical Fisher Information

Accessing Fisher information via the JVP as shown in equation 9 is straightforward, but it does not take advantage of any inherent structure of the diffused distribution. In this section, we initially derive the analytical Fisher information (AFI) of diffused distribution under a simplified setting, where

| Methods | Likelihood Evaluation | | | Adjoint Sampling | | |
|---|---|---|---|---|---|---|
| | Theoretical Time-cost | Practical Time-cost (s) | Theoretical Error Bound | Theoretical Time-cost | Practical Time-cost (s) | Approximation Error Bound |
| JVP | $c_1 d + c_2 d^2$ | 2195.48 | ✗ | $c_1 d + c_2 d$ | 0.155 | ✗ |
| AFI (**Ours**) | $2c_1 d$ | **0.072**$_{99\% \downarrow}$ | ✓(Proposition 7) | $2c_1 d$ | **0.063**$_{59\% \downarrow}$ | ✓(Proposition 8) |

Table 1: Comparison of Jacobian-vector-product (JVP) and our analytical-Fisher-information (AFI) in terms of per-iteration theoretical time-cost, practical time-cost, and approximation error bound. The theoretical time-cost is based on assumptions of a network access cost of $c_1 d$ time and a back-propagation on network cost of $c_2 d$ time ($c_2 \approx 4c_1$). The practical time-cost is tested using the SD-V1.5 model over 10k COCO prompts. Here, our JVP baseline calculates every element of the trace, and discussion on its approximation is deffer to Appendix C.5.

we assume that the initial $q_0$ is a sum of Dirac. Subsequently, we extend the AFI to a more general setting. Importantly, the AFI obtained in both settings does not involve any gradient calculations and is expressed into the initial data distribution, thus enabling the derivation of novel algorithms. Several studies Lu et al. (2022a); Benton et al. (2024) have investigated a similar form, but have not expressed it in terms of the initial data distribution. Our formulation can also be derived from a transformation of their formula. A detailed discussion on this topic is provided in Appendix C.4.

## 3.1 THE DIRAC SETTING

We start with a simple setting where we assume that the initial distribution is characterized as a sum of Dirac distributions composed of the set of samples in the dataset. If we suppose the dataset is denoted as $\{\boldsymbol{y}_i\}_{i=0}^N$, then the initial distribution follows

$$\text{(Dirac Setting)} \qquad q(\boldsymbol{x}, t)|_{t=0} = \frac{1}{N} \sum_{i=0}^N \delta(\boldsymbol{x} - \boldsymbol{y}_i), \qquad (10)$$

where exists a $0 < \mathcal{D}_y < \infty$ such that $\|\boldsymbol{y}_i\| \leq \mathcal{D}_y$ holds true for every $i$. In this Dirac setting, we derive the following *Analytical Fisher Information*, which is a weighted outer-product sum devoid of gradients and composed solely of the initial distribution and the noise schedule.

**Proposition 1.** *Defines $v_i(\boldsymbol{x}_t, t)$ as $\exp\left(-\frac{|\boldsymbol{x}_t - \alpha_t \boldsymbol{y}_i|^2}{2\sigma_t^2}\right) \in \mathbb{R}$ and $w_i(\boldsymbol{x}_t, t)$ as $\frac{v_i(\boldsymbol{x}_t, t)}{\sum_j v_j(\boldsymbol{x}_t, t)} \in \mathbb{R}$. If $q_0$ takes the form as in equation equation 10, the Fisher information matrix of the diffused distribution $q_t$ for $t \in (0, 1]$ can be analytically formulated as follows:*

$$\text{(Dirac AFI)} \qquad \boldsymbol{F}_t(\boldsymbol{x}_t, t) = \frac{1}{\sigma_t^2} \boldsymbol{I} - \frac{\alpha_t^2}{\sigma_t^4} \left[ \sum_i w_i \boldsymbol{y}_i \boldsymbol{y}_i^\top - \left( \sum_i w_i \boldsymbol{y}_i \right) \left( \sum_i w_i \boldsymbol{y}_i \right)^\top \right] \qquad (11)$$

*where we have simplified $w_i(\boldsymbol{x}_t, t)$ to $w_i$, as it does not lead to any confusion.*

We also find that the $\sum_i w_i \boldsymbol{y}_i$ component in equation 11 can be effectively approximated by the trained score network in the form of $y$-prediction, as demonstrated in the following proposition.

**Proposition 2.** *Given the diffusion training loss in equation 4, and if $q_0$ conforms to the form presented in equation 10, then the optimal $\bar{\boldsymbol{y}}_\theta(\boldsymbol{x}_t, t)$ can accurately estimate $\sum_i w_i \boldsymbol{y}_i$.*

## 3.2 THE GENERAL SETTING

We then begin to extend the AFI in equation 11 to a more general setting, where we only assume that the initial distribution $q_0$ is a measure on $\mathbb{R}^d$ with finite second momentum.

$$\text{(General Setting)} \qquad q_0 \in \mathcal{P}_2(\mathbb{R}^d). \qquad (12)$$

In this general setting, we derive the following *Analytical Fisher Information*, which is a weighted outer-product integral devoid of gradients.

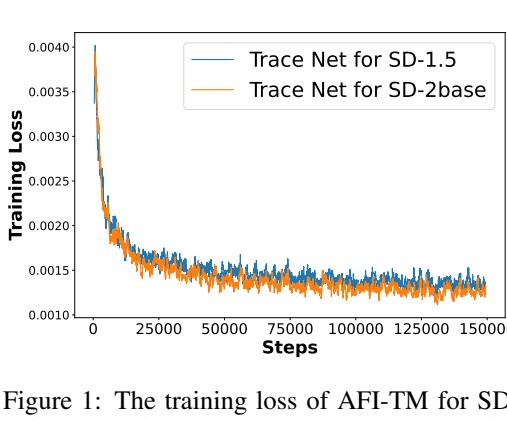

Figure 1: The training loss of AFI-TM for SD-1.5 and SD-2base. It demonstrates commendable convergence behavior.

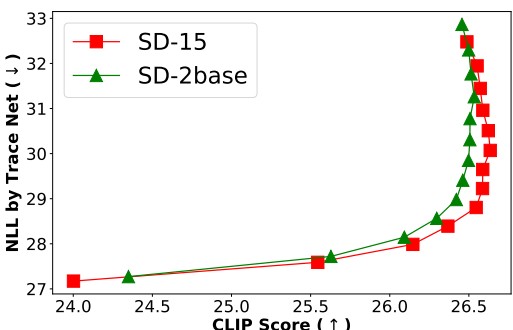

Figure 2: The trades-off curve of NLL and Clip score of SD-1.5 and SD-2base across various guidance scales in [1.5, 2.5, ..., 12.5, 13.5]

**Proposition 3.** *Let us define* $v(\boldsymbol{x}_t, t, \boldsymbol{y})$ *as* $\exp\left(-\frac{|\boldsymbol{x}_t - \alpha_t \boldsymbol{y}|^2}{2\sigma_t^2}\right) \in \mathbb{R}$ *and* $w(\boldsymbol{x}_t, t, \boldsymbol{y})$ *as* $\frac{v(\boldsymbol{x}_t, t, \boldsymbol{y})}{\int_{\mathbb{R}^d} v(\boldsymbol{x}_t, t, \boldsymbol{y}) \mathrm{d}q_0(\boldsymbol{y})} \in \mathbb{R}$. *If* $q_0$ *takes the form as in equation 12, the Fisher information matrix of the diffused distribution* $q_t$ *for* $t \in (0, 1]$ *can be analytically formulated as follows:*

$$(\textit{General AFI}) \quad \boldsymbol{F}_t(\boldsymbol{x}_t, t) = \frac{1}{\sigma_t^2}\boldsymbol{I} - \frac{\alpha_t^2}{\sigma_t^4}\left[\int w(\boldsymbol{y})\boldsymbol{y}\boldsymbol{y}^\top \mathrm{d}q_0 - \left(\int w(\boldsymbol{y})\boldsymbol{y}\mathrm{d}q_0\right)\left(\int w(\boldsymbol{y})\boldsymbol{y}\mathrm{d}q_0\right)^\top\right]$$

(13)

*where we simply write* $w(\boldsymbol{x}_t, t, \boldsymbol{y})$ *as* $w(\boldsymbol{y})$, *as long as it does not lead to any confusion.*

We further ascertain that the $\int w(\boldsymbol{y})\boldsymbol{y}\mathrm{d}q_0(\boldsymbol{y})$ component in equation 13 can be effectively approximated by the score network in the form of $y$-prediction, as demonstrated in the following proposition.

**Proposition 4.** *Given the diffusion loss in equation 4, and if* $q_0$ *conforms to the form in equation 12, then the optimal* $\bar{\boldsymbol{y}}_\theta(\boldsymbol{x}_t, t)$ *can accurately estimate* $\int w(\boldsymbol{y})\boldsymbol{y}\mathrm{d}q_0(\boldsymbol{y})$.

The derivation of the AFI under the general setting is akin to the sum in the Dirac setting but in an integral form. For the remainder of the paper, we will focus on developing our method based on the Dirac setting AFI. However, the same results can be naturally extended to the general setting.

## 4  AFI TRACE MATCHING (AFI-TM) METHOD

The likelihood evaluation of DMs would require access to Fisher Information's trace. In this section, we introduce a network to learn the trace, thus facilitating effective likelihood evaluation in DMs.

**Log-Likelihood in DMs**  Log-likelihood is a classic and significant metric for probabilistic generative models, extensively utilized for comparison between samples or models Bengio et al. (2013); Theis et al. (2015). According to Chen et al. (2018); Song et al. (2021), The log-likelihood of samples generated by PF-ODE in equation 6 from DMs can be computed through a connection to continuous normalizing flows as follows:

$$\begin{aligned}\frac{\partial \log q_t(\boldsymbol{x}_t, t)}{\partial t} &= -\mathrm{tr}\left(\frac{\partial}{\partial \boldsymbol{x}_t}\left(f(t)\boldsymbol{x}_t - \frac{1}{2}g^2(t)\partial_{\boldsymbol{x}_t}\log q_t(\boldsymbol{x}_t, t)\right)\right) \\ &= -\mathrm{tr}\left(\left(f(t)\boldsymbol{I} - \frac{1}{2}g^2(t)\frac{\partial^2}{\partial \boldsymbol{x}_t^2}\log q_t(\boldsymbol{x}_t, t)\right)\right) \\ &= -f(t)d - \frac{g^2(t)}{2}\mathrm{tr}\left(\boldsymbol{F}_t(\boldsymbol{x}_t, t)\right)\end{aligned}$$

(14)

where $\mathrm{tr}(\cdot)$ denotes the trace of a matrix, which is defined to be the sum of elements on the diagonal.

**Log-Likelihood Evaluation via JVP**    The current technique is only capable of conducting back-propagation of scalar value to the neural network. Therefore, the JVP in equation 9 cannot directly calculate the trace of the Fisher information. The JVP must iterate through each dimension to compute the individual elements on the diagonal, and then sum them up as follows

$$\text{(JVP for trace)} \qquad \text{tr}\left(\boldsymbol{F}_t(\boldsymbol{x}_t, t)\right) \approx \frac{1}{\sigma_t} \sum_{i=1}^{d} \frac{\partial \left[\left\langle \boldsymbol{\varepsilon}_\theta(\boldsymbol{x}_t, t) \middle| \boldsymbol{e}^{(i)} \right\rangle\right]}{\partial \boldsymbol{x}_t}. \tag{15}$$

Evaluating the trace using the JVP method would be extremely time-consuming due to the curse of dimensionality. If the time-complexity of a single backpropagation is $\mathcal{O}(d)$, then the calculation in equation 15 would have a time-complexity of $\mathcal{O}(d^2)$. In practice, as demonstrated in Table 1, evaluating the trace of Fisher information on the SD-1.5 model would require half an hour, rendering it nearly infeasible.

**Gradient-free Log-Likelihood Evaluation via AFI trace matching**    To overcome the limitations of the JVP method in evaluating the trace of the Fisher information, we propose to directly obtain its analytical form. Given the AFI in Proposition 1, we can also derive its trace in an analytical form of weighted norm sum, as highlighted in the following proposition:

> **Proposition 5.** *In the same context as Proposition 1, the trace of the Fisher information matrix for the diffused distribution $q_t$, where $t \in (0, 1]$, is given by:*
>
> $$\text{tr}\left(\boldsymbol{F}_t(\boldsymbol{x}_t, t)\right) = \frac{d}{\sigma_t^2} - \frac{\alpha_t^2}{\sigma_t^4}\left[\sum_i w_i \|\boldsymbol{y}_i\|^2 - \left\|\sum_i w_i \boldsymbol{y}_i\right\|^2\right] \tag{16}$$

As demonstrated in Proposition 2, the $\left\|\sum_i w_i \boldsymbol{y}_i\right\|^2$ can be directly estimated by $\|\bar{\boldsymbol{y}}_\theta(\boldsymbol{x}_t, t)\|^2$. Therefore, the only unknown element in equation 16 is $\sum_i w_i \|\boldsymbol{y}_i\|^2$. Consequently, we suggest estimating this term using a scalar-valued neural network, as per the following training algorithm:

---

**Algorithm 1** Training of AFI-TM Network

---

1: **Input**: data space dimension $d$, initial network $\boldsymbol{t}_\theta(\cdot, \cdot) : \mathbb{R}^d \times \mathbb{R} \mapsto \mathbb{R}$, noise schedule $\{\alpha_t\}$ and $\{\sigma_t\}$.
2: **repeat**
3: $\quad \boldsymbol{x}_0 \sim q_0\left(\boldsymbol{x}_0\right)$
4: $\quad t \sim \text{Uniform}(\{1, \ldots, T\})$
5: $\quad \boldsymbol{\varepsilon} \sim \mathcal{N}(\boldsymbol{0}, \mathbf{I})$
6: $\quad \boldsymbol{x}_t = \alpha_t \boldsymbol{x}_0 + \sigma_t \boldsymbol{\varepsilon}$
7: $\quad$ Take gradient descent step on $\nabla_\theta \left|\boldsymbol{t}_\theta(\boldsymbol{x}_t, t) - \frac{\|\boldsymbol{x}_0\|^2}{d}\right|^2$
8: **until** converged
9: **Output**: $\boldsymbol{t}_\theta(\cdot, \cdot)$

---

The training scheme detailed in Algorithm 1 can indeed enable $\boldsymbol{t}_\theta(\boldsymbol{x}_t, t)$ to estimate the weighted norm term $\frac{1}{d}\sum_i w_i(\boldsymbol{x}_t, t)\|\boldsymbol{y}_i\|^2$. This is substantiated by the convergence analysis Proposition 6, as presented below.

> **Proposition 6.** $\forall (x_t, t) \in \mathbb{R}^d \times \mathbb{R}_{\geq 0}$, *the optimal* $t_\theta(\boldsymbol{x}_t, t)$*s trained by the objective in Algorithm 1 are equal to* $\frac{1}{d}\sum_i w_i(\boldsymbol{x}_t, t)\|\boldsymbol{y}_i\|^2$.

Once we have obtained $\boldsymbol{t}_\theta$, we can evaluate the trace of Fisher Information in a gradient-free manner, as illustrated below. This approach is a straightforward result of equation 16 and Propositions 2 and 6, which we refer to as AFI trace matching (AFI-TM).

$$\text{(AFI-TM)} \qquad \text{tr}\left(\boldsymbol{F}_t(\boldsymbol{x}_t, t)\right) \approx d\left[\frac{1}{\sigma_t^2} - \frac{\alpha_t^2}{\sigma_t^4}\left(t_\theta(\boldsymbol{x}_t, t) - \left\|\frac{\boldsymbol{x}_t - \sigma_t \boldsymbol{\varepsilon}_\theta(\boldsymbol{x}_t, t)}{\alpha_t}\right\|^2\right)\right] \tag{17}$$

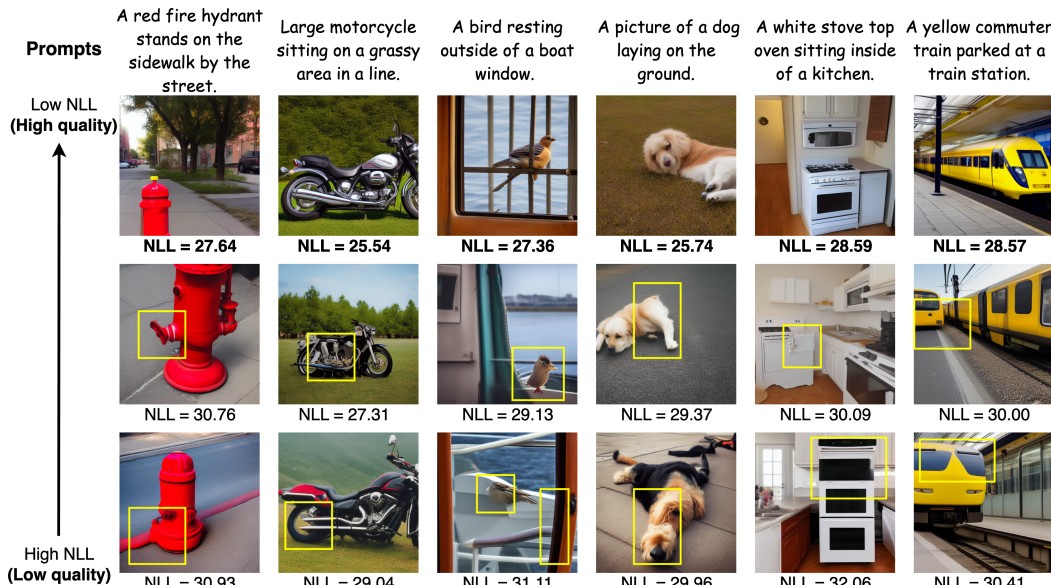

Figure 3: Our AFI-TM method facilitates the effective evaluation of the Negative Log-Likelihood (NLL) of generated samples with varying seeds. It can be demonstrated that a lower NLL signifies a region of higher possibility, thereby consistently indicating superior image quality.

To estimate the trace using AFI-TM in equation 17, we simply need one access to $t_\theta$ and $\varepsilon_\theta$. AFI-TM enables us to effectively evaluate the log-likelihood in a gradient-free manner with linear time complexity. We can further substantiate the theoretical approximation error bound when using AFI-TM to calculate the trace of the Fisher information, as illustrated in the Proposition 7.

**Proposition 7.** *Assume the approximation error on $t_\theta(\boldsymbol{x}_t, t)$ is $\delta_1$ and the approximation error on $\varepsilon_\theta(\boldsymbol{x}_t, t)$ is $\delta_2$, then the approximation error of the approximated Fisher trace equation 17 is at most $\frac{\alpha_t^2}{\sigma_t^4}\delta_1 + \frac{1}{\sigma_t^2}\delta_2^2$.*

**Experiments** We trained two AFI-TM networks for the SD-1.5 and SD-2base pipeline on the Laion2B-en dataset (Schuhmann et al., 2022), which contains 2.32 billion text-image pairs. Our $t_\theta$ follows the U-net structure, similar to stable diffusion models, but with an added MLP head to produce a scalar-valued output. We utilize the AdamW optimizer (Loshchilov & Hutter, 2019) with a learning rate of 1e-4. The training is executed across 8 Ascend 910B chips with a batch size of 384 and completes after 150K steps. In Figure 1, we demonstrate that the training loss of AFI-TM nets converges smoothly, indicating the robustness of the AFI-TM training scheme in Algorithm 1.

In Figure 2, we evaluate the average NLL and Clip score of samples generated by various SD models, using 10k randomly selected prompts from the COCO dataset Lin et al. (2014). A lower NLL suggests more realistic data generation, while a higher Clip score indicates a better match between the generated images and the input prompts. The results imply that the NLL and the Clip score form a trade-off curve across different guidance scales. This phenomenon, previously hypothesized in theory (Wu et al., 2024), is now confirmed in the SD models, thanks to the effective NLL evaluation via the AFI-TM method.

In Figure 3, we display images with varying NLL under the same prompt with 10 steps on SD-1.5 using DDIM. It's clear that images with lower NLL exhibit greater visual realism, while those with higher NLL often contain deformed elements (emphasized by the yellow rectangle). Our proposed NLL evaluation method proves to be an effective tool for automatic sample selection.

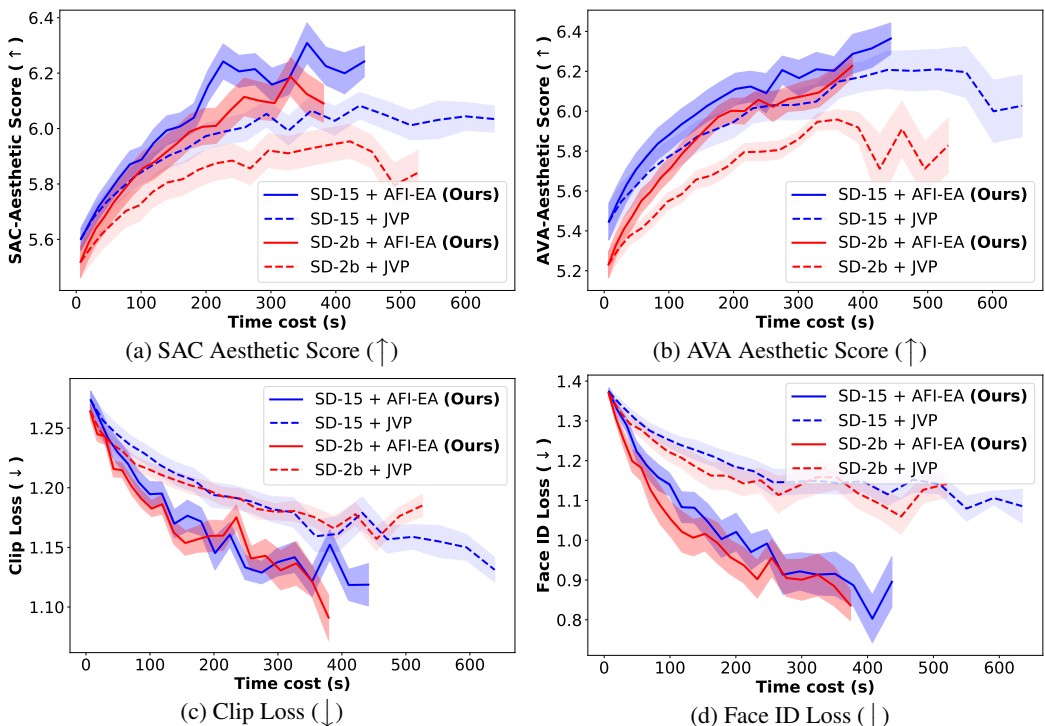

Figure 4: Comparison between our AFI method and the JVP method on adjoint guidance sampling across four objective scores: SAC/AVA aesthetic score, clip loss, and Face ID loss. Notably, our AFI consistently achieves superior scores with less time expenditure.

## 5 AFI ENDPOINT APPROXIMATION (AFI-EA) METHOD

The adjoint optimization of DMs would require applying Fisher Information as a linear operator. In this section, we present a training-free method that simplifies the complex linear transformation calculations thus enabling faster and more accurate adjoint optimization.

**Adjoint optimization sampling.** Guided sampling techniques are extensively utilized in diffusion models to facilitate controllable generation. Recently, to address the inflexibility of commonly used classifier-based guidance (Dhariwal & Nichol, 2021b) and classifier-free guidance (Ho & Salimans, 2022), a series of training-free adjoint guidance methods have been investigated and explored (Pan et al., 2023a;b).

Consider optimizing a scalar-valued loss function $\mathcal{L}(\cdot) : \mathbb{R}^d \mapsto \mathbb{R}$, which takes $\boldsymbol{x}_0$ in the data space as input. Adjoint guidance is implemented by applying gradient descent on $\boldsymbol{x}_t$ in the direction of $\frac{\partial \mathcal{L}(\boldsymbol{x}_0(\boldsymbol{x}_t))}{\partial \boldsymbol{x}_t}$. The essence of adjoint guidance is to use the gradient at $t = 0$ and follow the adjoint ODE (Pollini et al., 2018; Chen et al., 2018) to compute $\boldsymbol{\lambda}_t := \frac{\partial \mathcal{L}(\boldsymbol{x}_0(\boldsymbol{x}_t))}{\partial \boldsymbol{x}_t}$ for $t > 0$.

$$\text{(Adjoint ODE)} \qquad \frac{\mathrm{d}\boldsymbol{\lambda}_t}{\mathrm{d}t} = -\frac{\partial \boldsymbol{h}_\theta\left(\boldsymbol{x}_t, t\right)}{\partial \boldsymbol{x}_t}^\top \boldsymbol{\lambda}_t, \quad \boldsymbol{\lambda}_0 = \frac{\partial \mathcal{L}(\boldsymbol{x}_0)}{\partial \boldsymbol{x}_0} \qquad (18)$$

**Adjoint ODE via JVP.** Regardless of the ODE solver being used, it is necessary to compute the right-hand-side of equation 18, or equivalently, $\boldsymbol{F}(\boldsymbol{x}_t, t)^\top \boldsymbol{\lambda}_t$. This computation can be interpreted as applying the Fisher information matrix as a linear operator to the adjoint state $\boldsymbol{\lambda}_t$, from a functional analysis perspective (Yosida, 2012). Current practices utilize the JVP technique to approximate this linear transformation operation as follows:

$$\text{(JVP for Adjoint)} \qquad \boldsymbol{F}(\boldsymbol{x}_t, t)^\top \boldsymbol{\lambda}_t \approx \frac{1}{\sigma_t} \frac{\partial \boldsymbol{\varepsilon}_\theta\left(\boldsymbol{x}_t, t\right)}{\partial \boldsymbol{x}_t}^\top \boldsymbol{\lambda}_t \approx \frac{1}{\sigma_t} \frac{\partial \left[\langle \boldsymbol{\varepsilon}_\theta\left(\boldsymbol{x}_t, t\right) | \boldsymbol{\lambda}_t \rangle\right]}{\partial \boldsymbol{x}_t} \qquad (19)$$

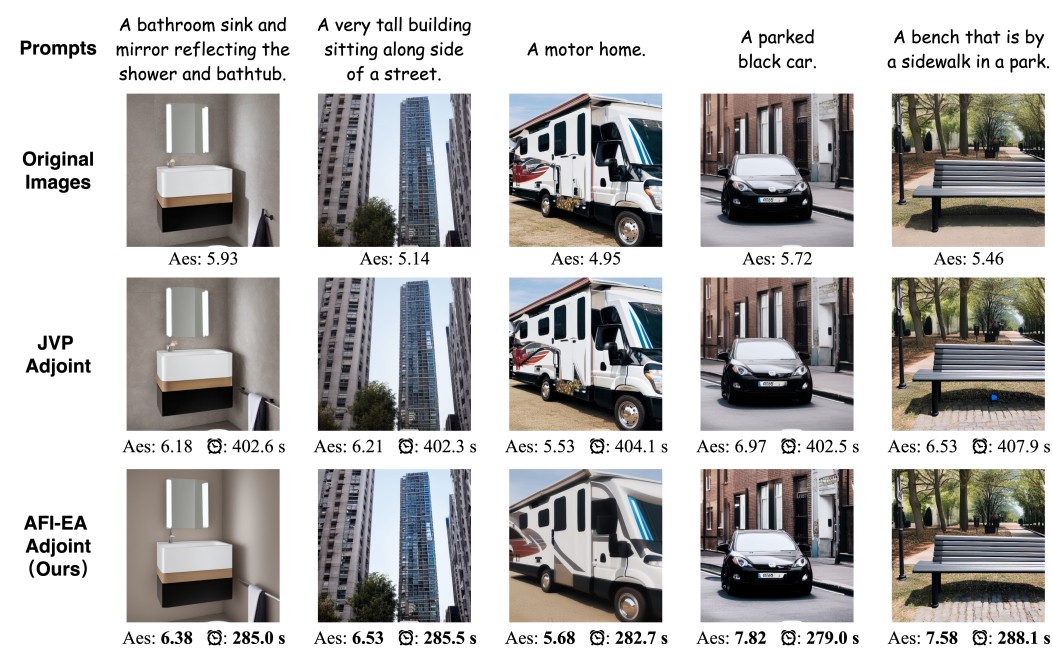

Figure 5: Qualitative comparison of AFI-EA (Ours) and JVP in the adjoint aesthetic improvement task. AFI-EA consistently generates high-aesthetic images with reduced time expenditure. The smoother visual effect is a desirable outcome of the target score, but not a result of our AFI-EA.

This process involves computationally intensive gradient operations on the neural network, and the approximation errors introduced by the JVP technique have no theoretical bound.

**Adjoint ODE via AFI-EA.** As previously discussed in Section 3, the AFI inherently doesn't require gradients, suggesting that we could potentially apply the Fisher information as a linear operator in a gradient-free manner. The challenging part in equation 11 is $\sum_i w_i \boldsymbol{y}_i \boldsymbol{y}_i^\top$, which represents a weighted form of outer-products of data. Based on the definition of $w_i$, the closest $\boldsymbol{y}_i$ to $\boldsymbol{x}_0$ will dominate as $t \to 0$. This makes it intuitive to replace this sum with a single final sample outer-product $\boldsymbol{x}_0 \boldsymbol{x}_0^\top$. It's also important to note that the adjoint guidance itself needs to compute $\boldsymbol{x}_0$ at each guidance step, eliminating the need for additional computation to obtain $\boldsymbol{x}_0$. Given that we utilize the endpoint sample $\boldsymbol{x}_0$, we refer to this approximation technique as AFI Endpoint Approximation (EA). The formulation for AFI-EA in adjoint ODE is as follows:

$$
\begin{aligned}
\text{(AFI-EA)} \quad \boldsymbol{F}(\boldsymbol{x}_t, t)^\top \boldsymbol{\lambda}_t &\approx \left( \frac{1}{\sigma_t^2} \boldsymbol{I} - \frac{\alpha_t^2}{\sigma_t^4} \left( \sum_i w_i \boldsymbol{y}_i \boldsymbol{y}_i^\top - \bar{\boldsymbol{y}}_\theta(\boldsymbol{x}_t, t) \bar{\boldsymbol{y}}_\theta(\boldsymbol{x}_t, t)^\top \right) \right)^\top \boldsymbol{\lambda}_t \\
&\approx \left( \frac{1}{\sigma_t^2} \boldsymbol{I} - \frac{\alpha_t^2}{\sigma_t^4} \left( \boldsymbol{x}_0 \boldsymbol{x}_0^\top - \bar{\boldsymbol{y}}_\theta(\boldsymbol{x}_t, t) \bar{\boldsymbol{y}}_\theta(\boldsymbol{x}_t, t)^\top \right) \right)^\top \boldsymbol{\lambda}_t \\
&= \frac{1}{\sigma_t^2} \boldsymbol{\lambda}_t - \frac{\alpha_t^2}{\sigma_t^4} \langle \boldsymbol{x}_0, \boldsymbol{\lambda}_t \rangle \boldsymbol{x}_0 + \frac{\alpha_t^2}{\sigma_t^4} \langle \bar{\boldsymbol{y}}_\theta(\boldsymbol{x}_t, t), \boldsymbol{\lambda}_t \rangle \bar{\boldsymbol{y}}_\theta(\boldsymbol{x}_t, t)
\end{aligned}
\tag{20}
$$

The AFI-EA approximation leads to a scalar-weighted combination of $\boldsymbol{\lambda}_t$, $\boldsymbol{x}_0$, and $\bar{\boldsymbol{y}}_\theta(\boldsymbol{x}_t, t)$, which importantly, does not involve any gradients. Additionally, we derive the theoretical approximation error bound of the AFI-EA in Proposition 8. To measure the accuracy of AFI-EA as a linear operator, we opt to use the Hilbert–Schmidt norm (Gohberg et al., 1990) for measurement, as follows:

**Proposition 8.** *Assume that the approximation error on $\boldsymbol{\varepsilon}_\theta(\boldsymbol{x}_t, t)$ is denoted as $\delta_2$, the approximation error of the endpoint approximated Fisher linear operator, as referenced in 20, is at most $\frac{\alpha_t^2}{\sigma_t^3} \left( 2\mathcal{D}_y^2 + \sqrt{d}\delta_2 \right)$ when measured in terms of the Hilbert–Schmidt norm.*

**Experiments on AFI-EA.** As depicted in Figure 4, we conducted experiments comparing our AFI-EA and JVP methods in adjoint guidance sampling, using four different scores and two different base models. AFI-EA consistently achieves better scores due to its bounded approximation error. Furthermore, AFI-EA requires less processing time as it eliminates the need for time-consuming gradient operations. AFI-EA and JVP are compared under the same guidance scales and schemes across various numbers of steps. Details regarding the score function can be found in Appendix B.2.

As depicted in Figure 5, our AFI-EA consistently generates samples with higher aesthetic scores with a reduced time-cost compared to JVP. It's worth noting that this enhancement in aesthetics results in final images that are more vibrant and smoother. All samples are generated within 50 steps, with adjoint applied from the 15th to the 35th step, details on hyperparameters can be found in Appendix B.2.

## 6 THEOREM ON THE OT PROPERTY OF THE PF-ODE DEDUCED MAP

There is an increasing trend towards analyzing the probability modeling capabilities of DMs by interpreting them from an optimal transport perspective (Albergo et al., 2023; Chen et al., 2024). The foundational concepts of optimal transport can be found in Appendix A.9. One of the central questions is whether the map deduced by the PF-ODE could represent an optimal transport. If so, how should we design the noise schedule, or what conditions should the data distribution meet? For a specific noise schedule where $f(t) \equiv 0$, Zhang et al. (2024a) demonstrates that having all data points lying on a single line is a sufficient condition for the map to represent an optimal transport.

In this section, we find out that the AFI, as derived in section 3, can contribute to the first equivalence condition for the OT property of PF-ODE deduced mapping under a general noise schedule. We refer to this theorem as the AFI Optimal Transport (AFI-OT) theorem. The AFI-OT condition is presented in the following:

---

**Theorem 1.** *Denote the diffeomorphism deduced by the PF-ODE 5 as follows*

$$T_{s,t} : \mathbb{R}^n \longrightarrow \mathbb{R}^n; \boldsymbol{x}_s \longmapsto \boldsymbol{x}_t, \quad \forall t \geq s > 0. \tag{21}$$

*This diffeomorphism is well-posed guaranteed by the global version of Picard-Lindelöf theorem Amann (2011); Zhang et al. (2024a). The diffeomorphism $T_{s,T}$ is a Monge optimal transport map if and only if the normalized fundamental matrix for $\boldsymbol{B}(t) \equiv \boldsymbol{B}(t, \boldsymbol{x}_t)$ at s is semi-positive definite for every PF-ODE chain start from a $\boldsymbol{x}_T \in \mathbb{R}^d$. where*

$$\boldsymbol{B}(t, \boldsymbol{x}_t) = \left[ f(t) - \frac{g^2(t)}{2\sigma_t^2} \right] \boldsymbol{I} + \frac{\alpha_t^2 g^2(t)}{2\sigma_t^4} \left[ \sum_i w_i \boldsymbol{y}_i \boldsymbol{y}_i^\top - \left( \sum_i w_i \boldsymbol{y}_i \right) \left( \sum_i w_i \boldsymbol{y}_i \right)^\top \right]. \tag{22}$$

---

The outline of the proof is as follows: We first apply Brenier's theorem (Brenier, 1991; Santambrogio, 2015) to convert the problem of whether the PF-ODE mapping is an optimal transport into the task of finding a convex potential, where the existence is guaranteed by the Poincaré's Theorem (Lang, 2012). We then use adjoint methods to express the second derivatives of the convex potential function in the form of an integral of information Rockafellar (2015); Pollini et al. (2018). Finally, we apply matrix exponential integration theory Masuyama (2016) to reformulate the condition into the AFI-OT theorem. Detailed proofs can be found in Appendix A.10.

## 7 CONCLUSIONS

This paper introduced the Analytical Fisher Information (AFI), an analytical formulation that allows for more efficient and theoretical exploration of Fisher Information of diffused distribution. Practically, we have proposed two algorithmic variants of AFI for different scenarios: AFI trace matching (AFI-TM) and AFI endpoint approximation (AFI-EA). Both methods are gradient-free, theoretically guaranteed for approximation error bounds and convergence properties, and offer improved accuracy and reduced time-cost compared to the traditional JVP method. Theoretically, we have established the first general theorem for the PF-ODE map to be optimal transport. This work not only improves the efficiency of Fisher Information evaluation but also widens our understanding of the diffused distributions. Please refer to further discussions in appendix C.

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

# Appendix

## Contents

## A  PROOFS AND FORMULATIONS

### A.1  PROOF OF PROPOSITION 1

Notice that, in the subsection, we can do an interchange of sum and gradient, this is due to the Leibniz's rule (Osler, 1970) and the boundness condition we set in equation 10. Before we give the proof of Proposition 1, we would like to establish two technical lemmas. The first lemma is about the first partial derivative of $v_i(\boldsymbol{x}_t, t)$ w.r.t. $\boldsymbol{x}_t$.

**Lemma 1.**

$$
\begin{aligned}
\frac{\partial v_i(\boldsymbol{x}_t, t)}{\partial \boldsymbol{x}_t} &= \frac{\partial \exp\left(-\frac{|\boldsymbol{x}_t - \alpha_t \boldsymbol{y}_i|^2}{2\sigma_t^2}\right)}{\partial \boldsymbol{x}_t} \\
&= -\frac{1}{\sigma_t^2}(\boldsymbol{x}_t - \alpha_t \boldsymbol{y}_i)\exp\left(-\frac{|\boldsymbol{x}_t - \alpha_t \boldsymbol{y}_i|^2}{2\sigma_t^2}\right) \\
&= -\frac{1}{\sigma_t^2}(\boldsymbol{x}_t - \alpha_t \boldsymbol{y}_i)v_i(\boldsymbol{x}_t, t)
\end{aligned}
\tag{23}
$$

The second lemma is about the Jacobian of $\sum_i w_i(\boldsymbol{x}_t, t)\,\boldsymbol{y}_i$ w.r.t. $\boldsymbol{x}_t$.

**Lemma 2.**

$$
\begin{aligned}
&\frac{\partial \sum_i w_i(\boldsymbol{x}_t, t)\,\boldsymbol{y}_i}{\partial \boldsymbol{x}_t} \\
&= \sum_i \boldsymbol{y}_i \left(\frac{\partial w_i(\boldsymbol{x}_t, t)}{\partial \boldsymbol{x}_t}\right)^\top \\
&= \sum_i \boldsymbol{y}_i \left(\frac{\partial}{\partial \boldsymbol{x}_t}\left[\frac{v_i(\boldsymbol{x}_t, t)}{\sum_j v_j(\boldsymbol{x}_t, t)}\right]\right)^\top \\
&= \sum_i \boldsymbol{y}_i \left\{\frac{\frac{\partial v_i(\boldsymbol{x}_t, t)}{\partial \boldsymbol{x}_t}\left[\sum_k v_k(\boldsymbol{x}_t, t)\right] - v_i(\boldsymbol{x}_t, t)\frac{\partial}{\partial \boldsymbol{x}_t}\left[\sum_j v_j(\boldsymbol{x}_t, t)\right]}{\left[\sum_k v_k(\boldsymbol{x}_t, t)\right]^2}\right\}^\top \\
&= \frac{1}{\left(\sum_k v_k\right)^2}\sum_i \boldsymbol{y}_i\left\{-\frac{1}{\sigma_t^2}(\boldsymbol{x}_t - \alpha_t \boldsymbol{y}_i)v_i\sum_k v_k - v_i(\boldsymbol{x}_t, t)\left(-\frac{1}{\sigma_t^2}\right)\sum_j (\boldsymbol{x}_t - \alpha_t \boldsymbol{y}_j)v_j\right\}^\top \\
&= \frac{1}{\left(\sum_k v_k\right)^2}\left(-\frac{1}{\sigma_t^2}\right)\sum_i v_i \boldsymbol{y}_i\left\{(\boldsymbol{x}_t - \alpha_t \boldsymbol{y}_i)\sum_k v_k - \sum_j (\boldsymbol{x}_t - \alpha_t \boldsymbol{y}_i)v_j\right\}^\top \\
&= \frac{1}{\left(\sum_k v_k\right)^2}\left(-\frac{1}{\sigma_t^2}\right)\sum_i v_i \boldsymbol{y}_i\left\{-\alpha_t \boldsymbol{y}_i\sum_k v_k + \alpha_t \sum_j \boldsymbol{y}_j v_j\right\}^\top \\
&= \frac{\alpha_t}{\sigma_t^2}\left[\frac{\sum_i v_i \boldsymbol{y}_i \boldsymbol{y}_i^\top (\sum_k v_k)}{\left(\sum_k v_k\right)^2} - \left(\frac{\sum_i v_i \boldsymbol{y}_i}{\sum_k v_k}\right)\left(\frac{\sum_i v_i \boldsymbol{y}_i}{\sum_k v_k}\right)^\top\right] \\
&= \frac{\alpha_t}{\sigma_t^2}\left[\sum_i w_i \boldsymbol{y}_i \boldsymbol{y}_i^\top - \left(\sum_i w_i \boldsymbol{y}_i\right)\left(\sum_i w_i \boldsymbol{y}_i\right)^\top\right]
\end{aligned}
\tag{24}
$$

Now we are ready to give the Proof of Proposition 1

*Proof.* According to the initial distribution (equation 10) and the diffusion kernel (equation 1), the marginal distribution at some time $t > 0$ would be

$$
p(\boldsymbol{x}_t, t) = \frac{1}{N}\sum_i \left(2\pi\sigma_t^2\right)^{-\frac{d}{2}}\exp\left(-\frac{|x_t - \alpha_t \boldsymbol{y}_i|^2}{2\sigma_t^2}\right)
\tag{25}
$$

Thus the log-density has the following analytical formulation

$$\log p(\boldsymbol{x}_t, t) = \log\left[\frac{1}{N}\left(2\pi\sigma_t^2\right)^{-\frac{d}{2}}\sum_i \exp\left(-\frac{|x_t - \alpha_t\boldsymbol{y}_i|^2}{2\sigma_{t^2}}\right)\right]$$

$$= \log\left[\sum_i \exp\left(-\frac{|x_t - \alpha_t\boldsymbol{y}_i|^2}{2\sigma_t^2}\right)\right] + C \qquad (26)$$

$$= \log\left[\sum_i v_i\left(\boldsymbol{x}_t, t\right)\right] + C$$

The score can be expressed as follows

$$\frac{\partial}{\partial\boldsymbol{x}_t}\log p\left(\boldsymbol{x}_t, t\right) = \frac{\partial}{\partial\boldsymbol{x}_t}\log\left[\sum_i v_i\left(\boldsymbol{x}_t, t\right)\right]$$

$$= \frac{\frac{\partial}{\partial\boldsymbol{x}_t}\left[\sum_i v_i\left(\boldsymbol{x}_t, t\right)\right]}{\sum_i v_i\left(\boldsymbol{x}_t, t\right)}$$

$$= \frac{-\frac{1}{\sigma_t^2}\sum_j\left(\boldsymbol{x}_t - \alpha_t y_j\right)v_j}{\sum_i v_i\left(\boldsymbol{x}_t, t\right)} \qquad (27)$$

$$= -\frac{1}{\sigma_t^2}\left[x_t - \alpha_t\sum_j w_j\left(\boldsymbol{x}_t, t\right)y_j\right]$$

The Fisher information we want can then be calculated by further applying a gradient on the score.

$$\boldsymbol{F}_t(\boldsymbol{x}_t, t) = -\frac{\partial}{\partial\boldsymbol{x}_t}\left(\frac{\partial}{\partial\boldsymbol{x}_t}\log p\left(\boldsymbol{x}_t, t\right)\right)$$

$$= -\frac{\partial}{\partial\boldsymbol{x}_t}\left\{-\frac{1}{\sigma_t^2}\left[x_t - \alpha_t\sum_j w_j\left(\boldsymbol{x}_t, t\right)y_j\right]\right\}$$

$$= \frac{1}{\sigma_t^2}\boldsymbol{I} - \frac{\alpha_t}{\sigma_t^2}\frac{\partial\sum_i w_i\left(\boldsymbol{x}_t, t\right)\boldsymbol{y}_i}{\partial\boldsymbol{x}_t} \qquad (28)$$

$$= \frac{1}{\sigma_t^2}\boldsymbol{I} - \frac{\alpha_t^2}{\sigma_t^4}\left[\sum_i w_i\boldsymbol{y}_i\boldsymbol{y}_i^\top - \left(\sum_i w_i\boldsymbol{y}_i\right)\left(\sum_i w_i\boldsymbol{y}_i\right)^\top\right] \quad \text{(by Lemma 2)}$$

$$\square$$

This proof is inherently the calculation of the Hessian of a log-convolution of a density, we provide the detailed derivation here for completeness.

## A.2 PROOF OF PROPOSITION 2

*Proof.* Given fixed $(x_t, t)$, $\mathcal{L}$ is a quadratic form of $y_\theta$. To obtain the optimal $\bar{\boldsymbol{y}}_\theta$, we differentiate $\mathcal{L}$ and set this derivative equal to zero, resulting in the following

$$0 = \frac{\partial\mathcal{L}}{\partial\bar{\boldsymbol{y}}_\theta(\boldsymbol{x}_t, t)} = \frac{\partial}{\partial\bar{\boldsymbol{y}}_\theta(\boldsymbol{x}_t, t)}\sum_j \underbrace{\frac{1}{N}(2\pi\sigma_t^2)^{-\frac{d}{2}}v_j(\boldsymbol{x}_t, t)\lambda_t}_{A_t}\frac{\alpha_t^2}{\sigma_t^2}\|\bar{\boldsymbol{y}}_\theta(\boldsymbol{x}_t, t) - \boldsymbol{y}_j\|^2$$

$$= 2A_t\lambda_t\frac{\alpha_t^2}{\sigma_t^2}\sum_j \boldsymbol{v}_j(x_t, t)(\bar{\boldsymbol{y}}_\theta(\boldsymbol{x}_t, t) - \boldsymbol{y}_j), \qquad (29)$$

which yields

$$\bar{\boldsymbol{y}}_\theta^*(\boldsymbol{x}_t, t) = \sum_k \frac{v_k(\boldsymbol{x}_t, t)}{\sum_j v_j(\boldsymbol{x}_t, t)} \boldsymbol{y}_k = \sum_i w_i \boldsymbol{y}_i. \tag{30}$$

$\square$

### A.3 PROOF OF PROPOSITION 3

Notice that, in the subsection, we can do an interchange of integral and gradient, this is due to the Leibniz's rule (Osler, 1970) and the bounded momentum condition we set in equation 12. Before we give the proof of Proposition 3, we would like to establish two technical lemmas. The first lemma is about the first partial derivative of $v(\boldsymbol{x}_t, t, \boldsymbol{y})$ w.r.t. $\boldsymbol{x}_t$.

**Lemma 3.**

$$\begin{aligned} \frac{\partial v(\boldsymbol{x}_t, t, \boldsymbol{y})}{\partial \boldsymbol{x}_t} &= \frac{\partial \exp\left(-\frac{|\boldsymbol{x}_t - \alpha_t \boldsymbol{y}|^2}{2\sigma_t^2}\right)}{\partial \boldsymbol{x}_t} \\ &= -\frac{1}{\sigma_t^2}(\boldsymbol{x}_t - \alpha_t \boldsymbol{y}) \exp\left(-\frac{|\boldsymbol{x}_t - \alpha_t \boldsymbol{y}|^2}{2\sigma_t^2}\right) \\ &= -\frac{1}{\sigma_t^2}(\boldsymbol{x}_t - \alpha_t \boldsymbol{y}) v(\boldsymbol{x}_t, t, \boldsymbol{y}) \end{aligned} \tag{31}$$

**Lemma 4.**

$$\begin{aligned} &\frac{\partial \int_{\mathbb{R}^d} w(\boldsymbol{x}_t, t, \boldsymbol{y}) \boldsymbol{y} \mathrm{d}q_0(\boldsymbol{y})}{\partial \boldsymbol{x}_t} \\ &= \int_{\mathbb{R}^d} \boldsymbol{y} \left(\frac{\partial w(\boldsymbol{x}_t, t, \boldsymbol{y})}{\partial \boldsymbol{x}_t}\right)^\top \mathrm{d}q_0(\boldsymbol{y}) \\ &= \int_{\mathbb{R}^d} \boldsymbol{y} \left(\frac{\partial}{\partial \boldsymbol{x}_t}\left[\frac{v(\boldsymbol{x}_t, t, \boldsymbol{y})}{\int_{\mathbb{R}^d} v(\boldsymbol{x}_t, t, \boldsymbol{y}') \mathrm{d}q_0(\boldsymbol{y}')}\right]\right)^\top \mathrm{d}q_0(\boldsymbol{y}) \\ &= \int_{\mathbb{R}^d} \boldsymbol{y} \left\{\frac{\frac{\partial v_i(\boldsymbol{x}_t, t)}{\partial \boldsymbol{x}_t}\left[\int_{\mathbb{R}^d} v(\boldsymbol{x}_t, t, \boldsymbol{y}') \mathrm{d}q_0(\boldsymbol{y}')\right] - v(\boldsymbol{x}_t, t, \boldsymbol{y})\frac{\partial}{\partial \boldsymbol{x}_t}\left[\int_{\mathbb{R}^d} v(\boldsymbol{x}_t, t, \boldsymbol{y}') \mathrm{d}q_0(\boldsymbol{y}')\right]}{\left[\int_{\mathbb{R}^d} v(\boldsymbol{x}_t, t, \boldsymbol{y}'') \mathrm{d}q_0(\boldsymbol{y}'')\right]^2}\right\}^\top \mathrm{d}q_0(\boldsymbol{y}) \\ &= \frac{1}{\left[\int v(\boldsymbol{y}'') \mathrm{d}q_0(\boldsymbol{y}'')\right]^2} \int \boldsymbol{y} \left\{-\frac{(\boldsymbol{x}_t - \alpha_t \boldsymbol{y}) v(\boldsymbol{y})}{\sigma_t^2} \int v(\boldsymbol{y}') \mathrm{d}q_0(\boldsymbol{y}') - v(\boldsymbol{y})\left(-\frac{1}{\sigma_t^2}\right) \int (\boldsymbol{x}_t - \alpha_t \boldsymbol{y}') v(\boldsymbol{y}') \mathrm{d}q_0(\boldsymbol{y}')\right\}^\top \mathrm{d}q_0(\boldsymbol{y}) \\ &= \frac{1}{\left[\int v(\boldsymbol{y}'') \mathrm{d}q_0(\boldsymbol{y}'')\right]^2}\left(-\frac{1}{\sigma_t^2}\right) \int v(\boldsymbol{y}) \boldsymbol{y} \left\{(\boldsymbol{x}_t - \alpha_t \boldsymbol{y}) \int v(\boldsymbol{y}') \mathrm{d}q_0(\boldsymbol{y}') - \int (\boldsymbol{x}_t - \alpha_t \boldsymbol{y}') v(\boldsymbol{y}') \mathrm{d}\boldsymbol{y}'\right\}^\top \mathrm{d}q_0(\boldsymbol{y}) \\ &= \frac{1}{\left[\int v(\boldsymbol{y}'') \mathrm{d}q_0(\boldsymbol{y}'')\right]^2}\left(-\frac{1}{\sigma_t^2}\right) \int v(\boldsymbol{y}) \boldsymbol{y} \left\{-\alpha_t \boldsymbol{y} \int v(\boldsymbol{y}') \mathrm{d}q_0(\boldsymbol{y}') + \alpha_t \int v(\boldsymbol{y}') \boldsymbol{y}' \mathrm{d}\boldsymbol{y}'\right\}^\top \mathrm{d}q_0(\boldsymbol{y}) \\ &= \frac{\alpha_t}{\sigma_t^2}\left[\frac{\int v(\boldsymbol{y}) \boldsymbol{y} \boldsymbol{y}^\top \mathrm{d}q_0(\boldsymbol{y}')\left[\int v(\boldsymbol{y}') \mathrm{d}q_0(\boldsymbol{y}')\right]}{\left[\int v(\boldsymbol{y}'') \mathrm{d}q_0(\boldsymbol{y}'')\right]^2} - \left(\frac{\int v(\boldsymbol{y}) \boldsymbol{y} \mathrm{d}q_0(\boldsymbol{y})}{\int v(\boldsymbol{y}'') \mathrm{d}q_0(\boldsymbol{y}'')}\right)\left(\frac{\int v(\boldsymbol{y}') \boldsymbol{y}' \mathrm{d}q_0(\boldsymbol{y}')}{\int v(\boldsymbol{y}'') \mathrm{d}q_0(\boldsymbol{y}'')}\right)^\top\right] \\ &= \frac{\alpha_t}{\sigma_t^2}\left[\int w(\boldsymbol{y}) \boldsymbol{y} \boldsymbol{y}^\top \mathrm{d}q_0(\boldsymbol{y}) - \left(\int w(\boldsymbol{y}) \boldsymbol{y} \mathrm{d}q_0(\boldsymbol{y})\right)\left(\int w(\boldsymbol{y}) \boldsymbol{y} \mathrm{d}q_0(\boldsymbol{y})\right)^\top\right] \end{aligned}$$

$$\tag{32}$$

*Proof.* According to the initial distribution (equation 12) and the diffusion kernel (equation 1), the marginal distribution at some time $t > 0$ would be

$$p(\boldsymbol{x}_t, t) = \int_{\mathbb{R}^d} \left(2\pi\sigma_t^2\right)^{-\frac{d}{2}} \exp\left(-\frac{|x_t - \alpha_t \boldsymbol{y}|^2}{2\sigma_{t^2}}\right) \mathrm{d}q_0(\boldsymbol{y}) \tag{33}$$

Thus the log-density has the following analytical formulation

$$\log q_t(\boldsymbol{x}_t, t) = \log \left[ \int_{\mathbb{R}^d} \left(2\pi\sigma_t^2\right)^{-\frac{d}{2}} \exp\left(-\frac{|x_t - \alpha_t \boldsymbol{y}|^2}{2\sigma_{t^2}}\right) \mathrm{d}q_0(\boldsymbol{y}) \right]$$

$$= \log \left[ \int_{\mathbb{R}^d} \exp\left(-\frac{|x_t - \alpha_t \boldsymbol{y}|^2}{2\sigma_{t^2}}\right) \mathrm{d}q_0(\boldsymbol{y}) \right] + C \tag{34}$$

$$= \log \left[ \int_{\mathbb{R}^d} v\left(\boldsymbol{x}_t, t, \boldsymbol{y}\right) \mathrm{d}q_0(\boldsymbol{y}) \right] + C$$

The score can be expressed as follows

$$\frac{\partial}{\partial \boldsymbol{x}_t} \log p\left(\boldsymbol{x}_t, t\right) = \frac{\partial}{\partial \boldsymbol{x}_t} \log \left[ \int_{\mathbb{R}^d} v\left(\boldsymbol{x}_t, t, \boldsymbol{y}\right) \mathrm{d}q_0(\boldsymbol{y}) \right]$$

$$= \frac{\frac{\partial}{\partial \boldsymbol{x}_t} \left[ \int_{\mathbb{R}^d} v\left(\boldsymbol{x}_t, t, \boldsymbol{y}\right) \mathrm{d}q_0(\boldsymbol{y}) \right]}{\int_{\mathbb{R}^d} v\left(\boldsymbol{x}_t, t, \boldsymbol{y}\right) \mathrm{d}q_0(\boldsymbol{y})}$$

$$= \frac{\int_{\mathbb{R}^d} \frac{\partial}{\partial \boldsymbol{x}_t} \left[ v\left(\boldsymbol{x}_t, t, \boldsymbol{y}\right) \right] \mathrm{d}q_0(\boldsymbol{y})}{\int_{\mathbb{R}^d} v\left(\boldsymbol{x}_t, t, \boldsymbol{y}\right) \mathrm{d}q_0(\boldsymbol{y})} \quad \text{(by xx and Leibniz integral rule)} \tag{35}$$

$$= \frac{-\frac{1}{\sigma_t^2} \int_{\mathbb{R}^d} (\boldsymbol{x}_t - \alpha_t \boldsymbol{y}) v\left(\boldsymbol{x}_t, t, \boldsymbol{y}\right) \mathrm{d}q_0(\boldsymbol{y})}{\int_{\mathbb{R}^d} v\left(\boldsymbol{x}_t, t, \boldsymbol{y}\right) \mathrm{d}q_0(\boldsymbol{y})}$$

$$= -\frac{1}{\sigma_t^2} \left[ x_t - \alpha_t \int_{\mathbb{R}^d} w(\boldsymbol{x}_t, t, \boldsymbol{y}) \boldsymbol{y} \mathrm{d}q_0(\boldsymbol{y}) \right]$$

The Fisher information we want can then be calculated by further applying a gradient on the score.

$$\boldsymbol{F}_t(\boldsymbol{x}_t, t) = -\frac{\partial}{\partial \boldsymbol{x}_t} \left( \frac{\partial}{\partial \boldsymbol{x}_t} \log p\left(\boldsymbol{x}_t, t\right) \right)$$

$$= -\frac{\partial}{\partial \boldsymbol{x}_t} \left\{ -\frac{1}{\sigma_t^2} \left[ x_t - \alpha_t \int_{\mathbb{R}^d} w(\boldsymbol{x}_t, t, \boldsymbol{y}) \boldsymbol{y} \mathrm{d}q_0(\boldsymbol{y}) \right] \right\}$$

$$= \frac{1}{\sigma_t^2} \boldsymbol{I} - \frac{\alpha_t}{\sigma_t^2} \frac{\partial \int_{\mathbb{R}^d} w(\boldsymbol{x}_t, t, \boldsymbol{y}) \boldsymbol{y} \mathrm{d}q_0(\boldsymbol{y})}{\partial \boldsymbol{x}_t}$$

$$= \frac{1}{\sigma_t^2} \boldsymbol{I} - \frac{\alpha_t^2}{\sigma_t^4} \left[ \int w_i \boldsymbol{y}\boldsymbol{y}^\top \mathrm{d}q_0(\boldsymbol{y}) - \left( \int w_i \boldsymbol{y} \mathrm{d}q_0(\boldsymbol{y}) \right) \left( \int w_i \boldsymbol{y} \mathrm{d}q_0(\boldsymbol{y}) \right)^\top \right] \quad \text{(by Lemma 4)}$$

$$\tag{36}$$

$$\square$$

This proof is also inherently the calculation of the Hessian of a log-convolution of a density, we provide the detailed derivation here for completeness.

### A.4 PROOF OF PROPOSITION 4

*Proof.* Given fixed $(x_t, t)$, $\mathcal{L}$ is a quadratic form of $y_\theta$. To obtain the optimal $\bar{y}_\theta$, we differentiate $\mathcal{L}$ and set this derivative equal to zero, resulting in the following

$$0 = \frac{\partial \mathcal{L}}{\partial \bar{\boldsymbol{y}}_\theta(\boldsymbol{x}_t, t)} = \frac{\partial}{\partial \bar{\boldsymbol{y}}_\theta(\boldsymbol{x}_t, t)} \int_{\mathbb{R}^d} \underbrace{\frac{1}{N} (2\pi\sigma_t^2)^{-\frac{d}{2}}}_{A_t} v(\boldsymbol{x}_t, t, \boldsymbol{y}) \lambda_t \frac{\alpha_t^2}{\sigma_t^2} \|\bar{\boldsymbol{y}}_\theta(\boldsymbol{x}_t, t) - \boldsymbol{y}\|^2 \mathrm{d}q_0(\boldsymbol{y})$$

$$= 2A_t \lambda_t \frac{\alpha_t^2}{\sigma_t^2} \int_{\mathbb{R}^d} \boldsymbol{v}(x_t, t, \boldsymbol{y})(\bar{\boldsymbol{y}}_\theta(\boldsymbol{x}_t, t) - \boldsymbol{y}) \mathrm{d}q_0(\boldsymbol{y}), \tag{37}$$

which yields

$$\bar{\boldsymbol{y}}_\theta^*(\boldsymbol{x}_t, t) = \int_{\mathbb{R}^d} \frac{v(\boldsymbol{x}_t, t, \boldsymbol{y}')}{\int_{\mathbb{R}^d} v(\boldsymbol{x}_t, t, \boldsymbol{y}'') \mathrm{d}q_0(\boldsymbol{y}'')} \boldsymbol{y}' \mathrm{d}q_0(\boldsymbol{y}') = \int_{\mathbb{R}^d} w(\boldsymbol{x}_t, t, \boldsymbol{y}') \boldsymbol{y}' \mathrm{d}q_0(\boldsymbol{y}'). \qquad (38)$$

$\square$

### A.5 PROOF OF PROPOSITION 5

**Lemma 5.** *Given a vector $\boldsymbol{v} \in \mathbb{R}^d$, the trace of the outer-product matrix of this vector is precisely equal to the square of its 2-norm. This can be shown as follows:*

$$\mathrm{tr}\left(\boldsymbol{v}\boldsymbol{v}^T\right) = \sum_{i=1}^d \left(\boldsymbol{v}\boldsymbol{v}^T\right)_{i,i} = \sum_{i=1}^d v_i * v_i = \|\boldsymbol{v}\|^2 \qquad (39)$$

*Proof.* Then we can start to give the derivation of Proposition 5

$$\begin{aligned}
\mathrm{tr}\left(\boldsymbol{F}_t(\boldsymbol{x}_t, t)\right) &= \mathrm{tr}\left(\frac{1}{\sigma_t^2}\boldsymbol{I} - \frac{\alpha_t^2}{\sigma_t^4}\left[\sum_i w_i \boldsymbol{y}_i \boldsymbol{y}_i^\top - \left(\sum_i w_i \boldsymbol{y}_i\right)\left(\sum_i w_i \boldsymbol{y}_i\right)^\top\right]\right) \\
&= \frac{1}{\sigma_t^2}\mathrm{tr}\left(\boldsymbol{I}\right) - \frac{\alpha_t^2}{\sigma_t^4}\left[\sum_i w_i \mathrm{tr}\left(\boldsymbol{y}_i \boldsymbol{y}_i^\top\right) - \mathrm{tr}\left(\left(\sum_i w_i \boldsymbol{y}_i\right)\left(\sum_i w_i \boldsymbol{y}_i\right)^\top\right)\right] \\
&= \frac{d}{\sigma_t^2} - \frac{\alpha_t^2}{\sigma_t^4}\left[\sum_i w_i \|\boldsymbol{y}_i\|^2 - \left\|\sum_i w_i \boldsymbol{y}_i\right\|^2\right]
\end{aligned} \qquad (40)$$

$\square$

### A.6 PROOF OF PROPOSITION 6

*Proof.* The objective in Algorithm 1 obviously equals to:

$$\arg\min_{t_\theta} \mathbb{E}_{\boldsymbol{x}_0 \sim q_0(\mathbf{x}_0), \boldsymbol{x}_t \sim \mathcal{N}(\alpha(t)\boldsymbol{x}_0, \sigma^2(t)\boldsymbol{I})} \left|\boldsymbol{t}_\theta(\boldsymbol{x}_t, t) - \frac{\|\boldsymbol{x}_0\|^2}{d}\right|^2. \qquad (41)$$

By expressing the expectation of Equation equation 41 in the form of a marginal distribution, we can transform the objective as follows:

$$\arg\min_{t_\theta} \sum_i \frac{1}{N}(2\pi\sigma_t^2)^{-\frac{d}{2}} \left|\boldsymbol{t}_\theta(\boldsymbol{x}_t, t) - \frac{\|\boldsymbol{y}_i\|^2}{d}\right|^2 \qquad (42)$$

The optimal $t_\theta^*$ must satisfy the condition that the gradient of the loss equals 0. Therefore, we have:

$$\begin{aligned}
0 &= \nabla_{t_\theta^*(\boldsymbol{x}_t, t)}\left[\sum_i \underbrace{\frac{1}{N}(2\pi\sigma_t^2)^{-\frac{d}{2}} v_i(\boldsymbol{x}_t, t)}_{A_t}\left|\frac{\|\boldsymbol{y}_i\|^2}{d} - t_\theta^*(\boldsymbol{x}_t, t)\right|^2\right] \\
&= \sum_i A_t v_i(\boldsymbol{x}_t, t)(t_\theta^*(\boldsymbol{x}_t, t) - \frac{\|\boldsymbol{y}_i\|^2}{d}) \\
&= A_t \sum_j v_j(\boldsymbol{x}_t, t) t_\theta^*(\boldsymbol{x}_t, t) - A_t \sum_i v_i(\boldsymbol{x}_t, t)\frac{\|\boldsymbol{y}_i\|^2}{d},
\end{aligned}$$

Thus

$$
\begin{aligned}
t_\theta^*(\boldsymbol{x}_t, t) &= \frac{A_t \sum_i v_i(\boldsymbol{x}_t, t) \frac{\|\boldsymbol{y}_i\|^2}{d}}{A_t \sum_j v_j(\boldsymbol{x}_t, t)} \\
&= \sum_i \frac{v_i(\boldsymbol{x}_t, t)}{\sum_j v_j(\boldsymbol{x}_t, t)} \frac{\|\boldsymbol{y}_i\|^2}{d} \\
&= \frac{1}{d} \sum_i w_i(\boldsymbol{x}_t, t) \|\boldsymbol{y}_i\|^2
\end{aligned}
\tag{43}
$$

We have successfully completed the proof that the optimal $t_\theta(\boldsymbol{x}_t, t)$, as trained by Algorithm 1, is equivalent to $\frac{1}{d} \sum_i w_i(\boldsymbol{x}_t, t) \|\boldsymbol{y}_i\|^2$. $\qquad\square$

## A.7 PROOF OF PROPOSITION 7

The approximation error of estimated trace equation 17 will be its difference from the true Fisher information trace equation 16. We use consecutive Cauchy–Schwartz and triangle inequality to get the bound of the approximation error:

$$
\begin{aligned}
&\left| \frac{d}{\sigma_t^2} - \frac{\alpha_t^2}{\sigma_t^4} \left[ \sum_i w_i \|\boldsymbol{y}_i\|^2 - \left\| \sum_i w_i \boldsymbol{y}_i \right\|^2 \right] - \left\{ d \left[ \frac{1}{\sigma_t^2} - \frac{\alpha_t^2}{\sigma_t^4} \left( t_\theta(\boldsymbol{x}_t, t) - \left\| \frac{\boldsymbol{x}_t - \sigma_t \boldsymbol{\varepsilon}_\theta(\boldsymbol{x}_t, t)}{\alpha_t} \right\|^2 \right) \right] \right\} \right| \\
=& \frac{\alpha_t^2}{\sigma_t^4} \left| \sum_i w_i \|\boldsymbol{y}_i\|^2 - \left\| \sum_i w_i \boldsymbol{y}_i \right\|^2 - \left( t_\theta(\boldsymbol{x}_t, t) - \left\| \frac{\boldsymbol{x}_t - \sigma_t \boldsymbol{\varepsilon}_\theta(\boldsymbol{x}_t, t)}{\alpha_t} \right\|^2 \right) \right| \\
\le& \frac{\alpha_t^2}{\sigma_t^4} \left[ \left| \sum_i w_i \|\boldsymbol{y}_i\|^2 - t_\theta(\boldsymbol{x}_t, t) \right| + \left\| \sum_i w_i \boldsymbol{y}_i - \frac{\boldsymbol{x}_t - \sigma_t \boldsymbol{\varepsilon}_\theta(\boldsymbol{x}_t, t)}{\alpha_t} \right\| \right] \\
\le& \frac{\alpha_t^2}{\sigma_t^4} \left[ \delta_1 + \frac{\sigma_t^2}{\alpha_t^2} \delta_2^2 \right] \\
=& \frac{\alpha_t^2}{\sigma_t^4} \delta_1 + \frac{1}{\sigma_t^2} \delta_2^2
\end{aligned}
\tag{44}
$$

## A.8 PROOF OF PROPOSITION 8

$$
\begin{aligned}
&\left\| \frac{1}{\sigma_t} \boldsymbol{I} - \frac{\alpha_t^2}{\sigma_t^3} \left[ \sum_i w_i \boldsymbol{y}_i \boldsymbol{y}_i^\top - \left( \sum_i w_i \boldsymbol{y}_i \right) \left( \sum_i w_i \boldsymbol{y}_i \right)^\top \right] - \left( \frac{1}{\sigma_t} \boldsymbol{I} - \frac{\alpha_t^2}{\sigma_t^3} \left( \boldsymbol{x}_0 \boldsymbol{x}_0^\top - \bar{\boldsymbol{y}}_\theta(\boldsymbol{x}_t, t) \bar{\boldsymbol{y}}_\theta(\boldsymbol{x}_t, t)^\top \right) \right) \right\|_{HS} \\
=& \left\| -\frac{\alpha_t^2}{\sigma_t^3} \left[ \sum_i w_i \boldsymbol{y}_i \boldsymbol{y}_i^\top - \left( \sum_i w_i \boldsymbol{y}_i \right) \left( \sum_i w_i \boldsymbol{y}_i \right)^\top \right] - \left( -\frac{\alpha_t^2}{\sigma_t^3} \left( \boldsymbol{x}_0 \boldsymbol{x}_0^\top - \bar{\boldsymbol{y}}_\theta(\boldsymbol{x}_t, t) \bar{\boldsymbol{y}}_\theta(\boldsymbol{x}_t, t)^\top \right) \right) \right\|_{HS} \\
\le& \frac{\alpha_t^2}{\sigma_t^3} \left\| \sum_i w_i \boldsymbol{y}_i \boldsymbol{y}_i^\top - \boldsymbol{x}_0 \boldsymbol{x}_0^\top \right\|_{HS} + \frac{\alpha_t^2}{\sigma_t^3} \left\| \left( \sum_i w_i \boldsymbol{y}_i \right) \left( \sum_i w_i \boldsymbol{y}_i \right)^\top - \bar{\boldsymbol{y}}_\theta(\boldsymbol{x}_t, t) \bar{\boldsymbol{y}}_\theta(\boldsymbol{x}_t, t)^\top \right\|_{HS} \\
=& \frac{\alpha_t^2}{\sigma_t^3} \sum_i w_i \left\| \boldsymbol{y}_i \boldsymbol{y}_i^\top - \boldsymbol{x}_0 \boldsymbol{x}_0^\top \right\|_{HS} + \frac{\alpha_t^2}{\sigma_t^3} \left\| \left( \sum_i w_i \boldsymbol{y}_i \right) \left( \sum_i w_i \boldsymbol{y}_i \right)^\top - \bar{\boldsymbol{y}}_\theta(\boldsymbol{x}_t, t) \bar{\boldsymbol{y}}_\theta(\boldsymbol{x}_t, t)^\top \right\|_{HS} \\
\le& \frac{\alpha_t^2}{\sigma_t^3} \sum_i w_i \max_i \left\| \boldsymbol{y}_i \boldsymbol{y}_i^\top - \boldsymbol{x}_0 \boldsymbol{x}_0^\top \right\|_{HS} + \frac{\alpha_t^2}{\sigma_t^3} \sum_j \left| \sum_i w_i \boldsymbol{y}_i[j] - \bar{\boldsymbol{y}}_\theta(\boldsymbol{x}_t, t)[j] \right| \left\| \sum_i w_i \boldsymbol{y}_i - \bar{\boldsymbol{y}}_\theta(\boldsymbol{x}_t, t) \right\| \\
\le& \frac{\alpha_t^2}{\sigma_t^3} \left( 2 \mathcal{D}_y^2 + \sqrt{d} \delta_2 \right)
\end{aligned}
\tag{45}
$$

### A.9 PRELIMINARIES ON OPTIMAL TRANSPORT

The optimal transport is the general problem of moving one distribution of mass to another as efficiently as possible. The *optimal transport problem* can be formulated in two primary ways, namely the Monge formulation (Monge, 1781) and the Kantorovich formulation (Kantorovich, 1960). Suppose there are two probability measures $\mu$ and $\nu$ on $(\mathbb{R}^n, \mathcal{B})$, and a cost function $c : \mathbb{R}^n \times \mathbb{R}^n \to [0, +\infty]$. The *Monge problem* is

$$\text{(MP)} \qquad \inf_{\mathrm{T}} \left\{ \int c(x, \mathrm{T}(x)) \, \mathrm{d}\mu(x) : \mathrm{T}_{\#}\mu = \nu \right\}. \tag{46}$$

The measure $\mathrm{T}_{\#}\mu$ is defined through $\mathrm{T}_{\#}\mu(A) = \mu(\mathrm{T}^{-1}(A))$ for every $A \in \mathcal{B}$ and is called the *pushforward* of $\mu$ through T.

It is evident that the Monge Problem (MP) transports the entire mass from a particular point, denoted as $x$, to a single point $\mathrm{T}(x)$. In contrast, Kantorovich provided a more general formulation, referred to as the *Kantorovich problem*:

$$\text{(KP)} \qquad \inf_{\gamma} \left\{ \int_{\mathbb{R}^n \times \mathbb{R}^n} c \, \mathrm{d}\gamma : \gamma \in \Pi(\mu, \nu) \right\}, \tag{47}$$

where $\Pi(\mu, \nu)$ is the set of *transport plans*, i.e.,

$$\Pi(\mu, \nu) = \left\{ \gamma \in \mathcal{P}(\mathbb{R}^n \times \mathbb{R}^n) : (\pi_x)_{\#}\gamma = \mu, (\pi_y)_{\#}\gamma = \nu \right\}, \tag{48}$$

where $\pi_x$ and $\pi_y$ are the two projections of $\mathbb{R}^n \times \mathbb{R}^n$ onto $\mathbb{R}^n$. For measures absolutely continuous with respect to the Lebesgue measure, these two problems are equivalent Villani et al. (2009). However, when the measures are discrete, they are entirely distinct as the constraint of the Monge Problem may never be fulfilled.

### A.10 PROOF OF THEOREM 1

To prove the Proposition 1, we first introduce two theorems to transform the problem of whether the PF-ODE mapping is a Monge map into the task of deciding the convexity of the potential function of $T_{s,T}$.

**Theorem 2.** *(Santambrogio, 2015, Theorem 1.48) Suppose that $\mu$ is a probability measure on $(\mathbb{R}^n, \mathcal{B})$ such that $\int |x|^2 \mathrm{d}\mu(x) < \infty$ and that $u : \mathbb{R}^n \to \mathbb{R} \cup \{+\infty\}$ is convex and differentiable $\mu$-a.e. Set $\mathrm{T} = \nabla u$ and suppose $\int |\mathrm{T}(x)|^2 \mathrm{d}\mu(x) < \infty$. Then $\mathrm{T}$ is optimal for the transport cost $c(x, y) = \frac{1}{2}|x - y|^2$ between the measures $\mu$ and $\nu = \mathrm{T}_{\#}\mu$.*

**Theorem 3.** ***The Brenier's Theorem.*** *(Santambrogio, 2015, Theorem 1.22) (Brenier, 1987; 1991) Let $\mu, v$ be probabilities over $\mathbb{R}^d$ and $c(x, y) = \frac{1}{2}|x - y|^2$. Suppose $\int |x|^2 \mathrm{d}x, \int |y|^2 \mathrm{d}y < +\infty$, which implies $\min(\text{KP}) < +\infty$ and suppose that $\mu$ gives no mass to $(d - 1)$ surfaces of class $C^2$. Then there exists, unique, an optimal transport map $\mathrm{T}$ from $\mu$ to $v$, and it is of the form $\mathrm{T} = \nabla u$ for a convex function $u$.*

To ensure the existence of the potential function, we need leverage the following

**Theorem 4.** ***The Poincaré's Theorem.*** *(Lang, 2012, Theorem 4.1 of Chapter V, §4) Let $U$ be an open ball in $\mathbb{R}^n$ and let $\omega$ be a differential form of degree $\geq 1$ on $U$ such that $\mathrm{d}\omega = 0$. Then there exists a differential form $\phi$ on $U$ such that $\mathrm{d}\phi = \omega$.*

**Remark 1.** *The conclusion remains valid when the open ball $U$ is substituted with the entirety of $\mathbb{R}^n$*

For $s > 0$, it is clear that $\frac{\mathrm{d}q_T(x_T)}{\mathrm{d}q_s(x_s)}$ is non-singular, and therefore, it satisfies the requirements of Brenier's Theorem 3, leading to the existence of a unique optimal transport map. According to

Theorem 2, if we can establish that the potential function of $T_{s,T}$ is convex, then the PF-ODE mapping will indeed be a Monge map. Notice that the existence of the potential map is guaranteed by Poincaré's Theorem 4.

We can now convert the condition of the potential function of $T_{s,T}$ being convex into the condition that its Jacobian, $\frac{\partial T_{s,T}(x_s)}{\partial x_s}$, is positive semi-definite, as per the following theorem in convex analysis.

> **Theorem 5.** *(Rockafellar, 2015, Theorem 4.5) Let $f$ be a twice continuously differentiable real-valued function on an open convex set $C$ in $R^n$. Then $f$ is convex on $C$ if and only if its Hessian matrix*
> $$Q_x = (q_{ij}(x)), \quad q_{ij}(x) = \frac{\partial^2 f}{\partial \xi_i \partial \xi_j}(\xi_1, \dots, \xi_n) \tag{49}$$
> *is positive semi-definite for every $x \in C$.*

If we denote that
$$\boldsymbol{A}(t) = \frac{\partial T_{t,T}(x_t)}{\partial x_t} \tag{50}$$

obviously, $\boldsymbol{A}(T) = I$ is p.s.d., our goal is to answer when $\boldsymbol{A}(t)$ is p.s.d.. We try to answer this to set up a connection between $\boldsymbol{A}(t)$ and $\boldsymbol{A}(T) = I$. We can derive that:

$$\frac{\mathrm{d}\boldsymbol{A}(t)}{\mathrm{d}t} = \lim_{\epsilon \to 0^+} \frac{\boldsymbol{A}(t+\epsilon) - \boldsymbol{A}(t)}{\epsilon}$$

$$= \lim_{\epsilon \to 0^+} \frac{\boldsymbol{A}(t+\epsilon) - \boldsymbol{A}(t+\epsilon)\nabla_{\boldsymbol{x}_t}\left[\boldsymbol{x}_t + \epsilon\left(f(t)\boldsymbol{x}_t - \frac{g^2(t)}{2}\nabla_{\boldsymbol{x}_t}\log_t q_t(\boldsymbol{x}_t)\right) + \mathcal{O}(\epsilon^2)\right]}{\epsilon}$$

$$= \lim_{\epsilon \to 0^+} \frac{\epsilon f(t)\boldsymbol{A}(t+\epsilon) + \frac{g^2(t)}{2}\epsilon\boldsymbol{A}(t+\epsilon)\nabla_{\boldsymbol{x}_t}\log_t q_t(\boldsymbol{x}_t) + \mathcal{O}(\epsilon^2)}{\epsilon}$$

$$= f(t)\boldsymbol{A}(t) + \frac{g^2(t)}{2}\boldsymbol{A}(t)\nabla_{\boldsymbol{x}_t}\log_t q_t(\boldsymbol{x}_t)$$

$$= f(t)\boldsymbol{A}(t) - \frac{g^2(t)}{2}\boldsymbol{A}(t)\left\{\frac{1}{\sigma_t^2}\boldsymbol{I} - \frac{\alpha_t^2}{\sigma_t^4}\left[\int w(\boldsymbol{y})\boldsymbol{y}\boldsymbol{y}^\top \mathrm{d}q_0(\boldsymbol{y}) - \left(\int w(\boldsymbol{y})\boldsymbol{y}\mathrm{d}q_0(\boldsymbol{y})\right)\left(\int w(\boldsymbol{y})\boldsymbol{y}\mathrm{d}q_0(\boldsymbol{y})\right)^\top\right]\right\}$$

$$= \boldsymbol{A}(t)\underbrace{\left\{\left[f(t) - \frac{g^2(t)}{2\sigma_t^2}\right]\boldsymbol{I} + \frac{\alpha_t^2 g^2(t)}{2\sigma_t^4}\left[\sum_i w_i\boldsymbol{y}_i\boldsymbol{y}_i^\top - \left(\sum_i w_i\boldsymbol{y}_i\right)\left(\sum_i w_i\boldsymbol{y}_i\right)^\top\right]\right\}}_{\boldsymbol{B}(t)}$$

$$\tag{51}$$

Notice that, the above ODE starts from $T$.

According to the Solution Matrices theory (Masuyama, 2016)[1], let us denote the $\boldsymbol{C}(t)$ is the normalized fundamental matrix at $T$ for $\boldsymbol{B}(T)$, which implies $C(t)$ is the solution to the following ODE:
$$C'(t) = B(t)C(t), C(T) = I, \quad \text{(flow from T to t)} \tag{52}$$

Then we can deduce that
$$\frac{\partial T_{t,T}(x_t)}{\partial x_t} = \boldsymbol{A}(t)$$
$$= \boldsymbol{C}(t)\boldsymbol{A}(T) \tag{53}$$
$$= \boldsymbol{C}(t)\boldsymbol{I}$$
$$= \boldsymbol{C}(t)$$

Thus the diffeomorphism $T_{s,T}$ is a Monge optimal transport map if and only if $\boldsymbol{C}(s)$ is semi-positive definite. Notice that the above requirement needs to be satisfied for every PF-ODE chain $x_t, t \in [T, t]$.

---

[1] The Definition 6.2 in `https://math.mit.edu/~jorloff/suppnotes/suppnotes03/ls6.pdf` suffice the result here.

## B    EXPERIMENTS DETAILS

### B.1    EVALUATION OF NLL

We employ the explicit Euler method to compute the NLL, excluding the final step near $t = 0$, for reasons discussed in Appendix C.1. We evaluate the NLL across 10 steps throughout the timeline of the PF-ODE. The AFI-TM network we trained uses float-point16 data type.

**Network architectures**    In terms of network architecture, we employ an SD Unet structure with an additional MLP head. However, we believe that a lighter network could potentially be sufficient for AFI-TM.

**Training cost of AFI-TM**    For training the AFI-TM network, we approximately spend 24 hours using 8 Ascend 910B chips. Given the size of the Laion2B-en dataset (which contains 2.32 billion images), this is quite an efficient speed. Additionally, the convergence behavior of the training loss is robust, as illustrated in Figure 1. We also hypothesize that our network design has redundancy, suggesting that we could further reduce costs by opting for lighter networks. We will provide more details about training costs in our revised manuscript.

### B.2    ADJOINT GUIDANCE SAMPLING

For Figure 4, we examined varying numbers of adjoint guidance, ranging from 0 to 20, under a full inference number of 50. The adjoint guidance scale was grid-searched by the JVP method.

**Base method for AFI-EA**    Several variants of adjoint-optimization algorithms exist, such as AdjointDPM (Pan et al., 2023a), AdjointDES (Blasingame & Liu, 2024), and SAG (Pan et al., 2023b). However, while these algorithms differ in their design of solvers for the adjoint ODE, they all utilize JVP when accessing Fisher information. We selected SAG as our base method due to its state-of-the-art performance. We believe that replacing JVP with AFI-EA could also enhance the performance of algorithms like AdjointDPM and AdjointDES.

**The Design of Score functions**

- Aesthetic Score
  For aesthetic score predictor $f_{aes} : \mathbb{R}^d \mapsto \mathbb{R}$, the adjoint optimization target is simply $f_{aes}$ itself.
  $$\mathcal{L}(\boldsymbol{x}_0) := f_{aes}(\boldsymbol{x}_0) \tag{54}$$

- Clip Loss
  Following the implementation of (Pan et al., 2023a), we use the features from the CLIP image encoder as our feature vector. The loss function is $L_2$-norm between the Gram matrix of the style image and the Gram matrix of the estimated clean image.
  $$\mathcal{L}(\boldsymbol{x}_0) := \left\| \text{clip}(\boldsymbol{x}_0)\text{clip}(\boldsymbol{x}_0)^\top - \text{clip}(\boldsymbol{x}_{ref})\text{clip}(\boldsymbol{x}_{ref})^\top \right\| \tag{55}$$

- FaceID Loss
  Following the implementation of (Pan et al., 2023b), we use ArcFace to extract the target features of reference faces to represent face IDs and compute the $l_2$ Euclidean distance between the extracted ID features of the estimated clean image and the reference face image as the loss function.
  $$\mathcal{L}(\boldsymbol{x}_0) := \|\text{ArcFace}(\boldsymbol{x}_0) - \text{ArcFace}(\boldsymbol{x}_{ref})\| \tag{56}$$

**Hyperparameters**    For the hyperparameters in adjoint guided sampling, we ensure a fair comparison between the JVP and AFI-EA methods. For most hyperparameters, we directly adopt the settings from previous works Pan et al. (2023b) for both the baseline JVP method and our AFI-EA method. For the guidance scale, we tune the value for the JVP method and use the same value for

our AFI-EA method. The AFI-EA method does not introduce additional hyperparameters, and for mutual hyperparameters, our AFI-EA method uses the exact same values as the JVP method. We adopt this strategy because our AFI-EA method solely improves the approximation of the Fisher information linear operator, without altering the adjoint sampling mechanism. Therefore, the suitable hyperparameters should remain unchanged, and we simply use the same parameters from the JVP method for our AFI-EA method. For all experiments, we set the number of sampling steps to $T = 50$. Adjoint guidance is applied starting from steps ranging from 15 to 35 and ending at step 35, with one guidance per step. Thus the only parameter we tune is the guidance strength. We determine this value for the JVP method via a grid search from 0.1 to 0.5 with a step size of 0.1 and find that the optimal guidance strength for JVP is 0.2. We then use this value for our AFI-EA method. Notice that, we apply a normalization to the guidance gradient for both JVP and AFI-EA methods, making our optimal guidance strength consistent across different scores. The tuning is conducted on 1k COCO prompts, and the computational budget for tuning is 4 * 5 * 3 GPU hours (4 tasks * 5 grids * 3 hours per single test) on Ascend 910B chips.

### B.3    PRETRAINED MODELS

All of the pretrained models used in our research are open-sourced and available online as follows:

- stable-diffusion-v1-5

  `https://huggingface.co/runwayml/stable-diffusion-v1-5`

- stable-diffusion-2-base

  `https://huggingface.co/stabilityai/stable-diffusion-2-base`

- SAC-aesthetic score predictor

  `https://github.com/christophschuhmann/improved-aesthetic-predictor/blob/main/sac%2Blogos%2Bava1-l14-linearMSE.pth`

- AVA-aesthetic score predictor

  `https://github.com/christophschuhmann/improved-aesthetic-predictor/blob/main/ava%2Blogos-l14-linearMSE.pth`

- ArcFace ID loss

  `https://github.com/TreB1eN/InsightFace_Pytorch`

- Clip loss

  `https://huggingface.co/openai/clip-vit-large-patch14`

## C    DISCUSSIONS

### C.1    SINGULARITY OF FISHER INFORMATION AT $t = 0$

Previous studies (Yang et al., 2023; Zhang et al., 2024b) have shown that the diffusion model, particularly when learned in $\epsilon$-prediction, can encounter a singularity issue at $t = 0$. Our AFI in equation 11 reaffirms this issue, as this formulation becomes ill-formed at $t = 0$ due to division by zero ($\sigma_0$). Consequently, our formulation does not describe the behavior at $t = 0$. The deep theoretical exploration of the singularity problem remains an open question in the diffusion model field. However, as it is not the primary focus of this paper, we will not discuss the AFI at $t = 0$.

### C.2    STATISTICAL CALIBER OF NEGATIVE LOG-LIKELIHOOD

When dealing with high-dimensional data such as images, direct likelihood comparisons may encounter scaling issues due to the dimensionality. In this study, unless explicitly indicated otherwise, we adopt the approach of Zheng et al. (2023) and typically use Negative Log-Likelihood (NLL) to refer to Bits Per Dimension (BPD).

$$\text{BPD} = \mathbb{E}_{\boldsymbol{x}_0 \sim q_0} \left[ \frac{-\log P_0\left(\boldsymbol{x}_0\right)}{d \log 2} \right] \tag{57}$$

## C.3 THE ODE SOLVERS

To compute the numerical solutions for the PF-ODE in equation 5, the likelihood ODE in equation 14, and the adjoint ODE in equation 18, we require ODE solvers. In our paper's experiments, we consistently use the explicit Euler method (referred to as DDIM when applied to PF-ODE). However, it's important to note that our approach is not dependent on a specific ODE solver. We can also utilize alternatives like fast ODE solvers (Lu et al., 2022b;c; Liu et al., 2022) or exact inversion ODE solvers (Wallace et al., 2023; Zhang et al., 2023).

## C.4 THE RELATION OF AFI TO COVARIANCE IN DMs

We note that there is a series of studies aiming to learn the covariance of the reverse diffusion Stochastic Differential Equation (SDE) (Bao et al., 2022b;a). In Bayesian statistics, Fisher information is defined as the covariance of the score. These studies derive their formulation by analyzing the covariance of the score and obtaining the Fisher information in terms of the score. However, our AFI is derived directly from the marginal distribution and is composed solely of the initial distribution and noise schedule. Furthermore, our application is unique; we are the first to replace the use of JVP with AFI, while the focus of these studies is to enhance the performance of Diffusion Models (DMs) with analytical covariance.

A very similar form of the Fisher information is proposed in Lemma 5 of Benton et al. (2024). The difference is that our Proposition 3 presents the specific form of Fisher information in terms of data distribution, which is not included in Lemma 5 of Benton et al. (2024). This distinction is crucial as it facilitates the derivation of our new algorithms. Also, we adopt the currently more commonly used $\alpha_t, \sigma_t$ notation to represent the noise schedule. Instead, Benton et al. (2024) $\alpha_t \equiv e^{-2t}$. We gave out this detailed formulation to avoid unnecessary misunderstandings in the development of the subsequent training scheme.

## C.5 COMPARISON OF AFI-TM TO HUTCHINSON TRACE ESTIMATOR

We notice that the naive trace calculation of the JVP method can be accelerated by the Hutchinson trace estimator. We have not conducted experiments using the Hutchinson estimator, as it is a Monte-Carlo type estimation, unlike the direct evaluation methods like the full-JVP baseline and our AFI-TM. Furthermore, the JVP method under the help of the Hutchinson method and its variants is still considerably more expensive than our method, and their applicability is also more restricted.

- **The Hutchinson method is much more costly:** To attain a relative error less than $\epsilon$ with a probability of $1 - \delta$, the Hutchinson method requires $\frac{2(1 - \frac{8}{3}\epsilon)\log\frac{1}{\delta}}{\epsilon^2}$ samples Skorski (2021). Assuming that the goal is to obtain an estimation with a relative error of less than 10% with a probability exceeding 90%, a minimum of 351 NFEs is required. This equates to 1 or 2 minutes on SD-v1.5 for a single trace estimation. For a complete NLL estimation of an image with 20 steps, this would take 20 minutes, which is entirely impractical in any business context. In contrast, our AFI-TM only requires 1 NFE for a single trace estimation, needing merely 10 seconds for the full NLL estimation of an image.

- **The application of the Hutchinson method is more restricted:** Due to its Monte-Carlo characteristics, the Hutchinson method is more appropriate for contributing to the computation of certain training objectives as in Lu et al. (2022a), where the unbiased property is sufficient, and large variance may be absorbed into network training. However, the Hutchinson method may encounter difficulties with accurate per-sample trace estimation. Our method can accommodate both scenarios, including per-sample computation.

There are already attempts to use the Hutchinson method to expedite the JVP of trace estimation in diffusion models Lu et al. (2022a). Nonetheless, due to Hutchinson's limitations, these practices are restricted to relatively small DMs (CIFAR-10 at most). We will add discussions on the Hutchinson method in our revision.

## C.6 DISCUSSIONS ON THE THEORETICAL BOUND OF AFI-EA

We notice that the error bound in Proposition 8 does not vanish as the training error decreases. From the rigorous theoretical perspective, considering the error bounds in Propositions 7 and 8, the AFI-EA method is less valid than the AFI-TM method, as we currently cannot establish vanishing bounds for it. However, from an empirical perspective, our AFI-EA outperforms the naive JVP method in terms of score improvement across various tasks and pretrained models, as demonstrated in Figure 4. Thus, the AFI-EA approximation proves to be valid in a practical sense. The replacement $\sum_i \boldsymbol{y}_i \boldsymbol{y}_i^\top \approx \boldsymbol{x}_0 \boldsymbol{x}_0^\top$ originates from the observation that $w_i$ is a weighting of the summation equal to 1, and as $t$ approaches 0, the $w_i$ closest to $\boldsymbol{x}_0$ will dominate due to the diffusion kernel. Therefore, this approximation is intuitively reasonable near $t = 0$, which is precisely where we apply adjoint guidance.

## C.7 BROADER (SOCIAL) IMPACTS

The development of accurate Fisher information of diffused distribution, as discussed in this paper, holds significant potential for several domains, including machine learning, healthcare, environmental modeling, and economics.

However, while this research holds great potential for positive impacts, it is also important to consider potential negative societal impacts. The enhanced ability of generative models given by AFI could potentially be misused. For instance, it could be exploited to create aesthetic-improved deepfakes, leading to misinformation. In healthcare, if not properly regulated, the use of synthetic patient data could lead to ethical issues. Therefore, it is crucial to ensure that the findings of this research are applied ethically and responsibly, with necessary safeguards in place to prevent misuse and protect privacy.

## C.8 LIMITATIONS

This paper does not explore the integration of AFI into accelerated ODE solvers. This paper is constrained in the scope of DMs, but similar second-order information may also exist in flow matching generative models. The AFI may also contribute to a more effective inference method, which we did not explore.

