# OpenReview forum: "Gradient-Free Analytical Fisher Information of Diffused Distributions"
_ICLR.cc/2025/Conference — Submitted to ICLR 2025_

### Official Review · Reviewer_iHvF · 2024-10-23

**Soundness:** 3
**Presentation:** 3
**Contribution:** 3
**Rating:** 6
**Confidence:** 2

**Summary:**

A method for efficiently evaluating Fisher information is developed for diffusion models. The method is based on its analytic formula, which is derived by consecutive differentials to the diffused distributions. Due to the Gaussian nature of the diffusion process, the formula is expressed as a weighted sum of outer-products of the score and initial data, which is free from computing gradients. Based on the formula, an efficient algorithm for assessing trace of Fisher information matrix is developed using neural network learning. Another efficient algorithm is also proposed for using Fisher information as a linear operator for the adjoint optimization of  diffusion models. Several theoretical guarantees are provided for the algorithms. The analytical knowledge on Fisher information is also utilized for deriving a theorem that allows mapping deduced by the probability flow ordinary differential equation to possess the optimal transportation property.

**Strengths:**

Computing Fisher information matrix for high dimensional distributions is computationally very hard. This paper develops efficient methods to compute its summarized quantities without directly computing the matrix. Not only that, the authors manage to provide a theorem that guarantees the optimal transportation property of the probability flow ODE utilizing the analytical knowledge of the Fisher information.

**Weaknesses:**

Training cost of AFI-TM is not fully described. As for AFI-EA, adjoint guided sampling generally requires hyper-parameters. Since how the parameters were tuned is not written, it is unclear whether the obtained gain in the performance is actually due to AFI-EA or not.

**Questions:**

1. What are the minimum computational requirements for training AFI-TM?
2. What was the total wall-clock training time for AFI-TM?
3. For the adjoint guided sampling experiments, please provide details on:
    i) Specific hyperparameters tuned,
   ii) Hyperparameter ranges explored,
  iii) Hyperparameter optimization method used,
  iv) Computational budget for tuning

---

> ### Author Response · Authors · 2024-11-20
> **Reply to Reviewer iHvF**
>
> Thank you for your valuable reviews, we will answer your questions one by one regarding these weaknesses/questions.
>
> > **Answer to Weaknesses 1:**  Training cost of AFI-TM is not fully described.
>
> For training the AFI-TM network, we approximately spend 24 hours using 8 Ascend 910B chips. Given the size of the Laion2B-en dataset (which contains 2.32 billion images), this is quite an efficient speed. Additionally, the convergence behavior of the training loss is robust, as illustrated in Figure 1. We also hypothesize that our network design has redundancy, suggesting that we could further reduce costs by opting for lighter networks. We will provide more details about training costs in our revised manuscript.
>
> > **Answer to Weaknesses 1:**  As for AFI-EA, adjoint guided sampling generally requires hyper-parameters. Since how the parameters were tuned is not written, it is unclear whether the obtained gain in the performance is actually due to AFI-EA or not.
>
> For the hyperparameters in adjoint guided sampling, we ensure a fair comparison between the JVP and AFI-EA methods. For most hyperparameters, we directly adopt the settings from previous works [1] for both the baseline JVP method and our AFI-EA method. For the guidance scale, we tune the value for the JVP method and use the same value for our AFI-EA method. The AFI-EA method does not introduce additional hyperparameters, and for mutual hyperparameters, our AFI-EA method uses the exact same values as the JVP method. We adopt this strategy because our AFI-EA method solely improves the approximation of the Fisher information linear operator, without altering the adjoint sampling mechanism. Therefore, the suitable hyperparameters should remain unchanged, and we simply use the same parameters from the JVP method for our AFI-EA method.
>
> [1] Jiachun Pan, Hanshu Yan, Jun Hao Liew, Jiashi Feng, and Vincent YF Tan. Towards accurate guided diffusion sampling through symplectic adjoint method. arXiv preprint arXiv:2312.12030, 2023.
>
> > **Answer to Questions 1:** What are the minimum computational requirements for training AFI-TM?
>
> Our trace network design adopts the U-net structure, similar to stable diffusion models, with only a negligible MLP head. Our training process approximately utilizes 9GB of CUDA memory. As such, our training can even be executed on an NVIDIA RTX 3090.
>
> > **Answer to Questions 2:** What was the total wall-clock training time for AFI-TM?
>
> To train the AFI-TM network, we approximately spend 24 hours using 8 Ascend 910B chips, completing 150K steps in the process.
>
> > **Answer to Questions 3:** For the adjoint guided sampling experiments, please provide details on: i) Specific hyperparameters tuned, ii) Hyperparameter ranges explored, iii) Hyperparameter optimization method used, iv) Computational budget for tuning
>
> For all experiments, we set the number of sampling steps to $T=50$. Adjoint guidance is applied starting from steps ranging from 15 to 35, and ending at step 35, with one guidance per step. Thus:
>
> **i)**  The only parameter we tune is the guidance strength.
>
> **ii)** and **iii)**  We determine this value for the JVP method via a grid search from 0.1 to 0.5 with a step size of 0.1, and find that the optimal guidance strength for JVP is 0.2. We then use this value for our AFI-EA method. Notice that, we apply a normalization to the guidance gradient for both JVP and AFI-EA methods, making our optimal guidance strength is consistent across different scores.
>
> **iv)** The tuning is conducted on 1k COCO prompts, and the computational budget for tuning is 4 * 5 * 3 GPU hours (4 tasks * 5 grids * 3 hours per single test) on Ascend 910B chips.
>
> We will provide additional details regarding hyperparameter tuning and settings in our revised manuscript.

---

> > ### Author Response · Authors · 2024-11-25
> > **Rebuttal Revision**
> >
> > Dear Reviewer iHvF,
> >
> > We have revised our manuscript with your valuable suggestions and feedback. The revised version has been uploaded. In this updated version, we have endeavored to address all the points you raised in your review. In particular:
> >
> > 1. We provided a comprehensive discussion of the training costs of our AFI-TM in Appendix B.1.
> > 2. We also provide a comprehensive description of our hyperparameter tunning of adjoint guidance experiments in Appendix B.2.
> >
> > All of our revised parts have been outlined in blue color. Thanks again for your helpful feedback!

---

### Official Review · Reviewer_jSGi · 2024-10-31

**Soundness:** 3
**Presentation:** 3
**Contribution:** 2
**Rating:** 6
**Confidence:** 3

**Summary:**

In this paper,
1. the authors derive exact form of the Fisher Information (FI) matrix of the diffused target distribution in diffusion models context.
2. Based on the formula, they propose an estimation of FI's trace by the trained diffusion model and an additionally trained model;
    - they also analyze its estimation errors,
    - they demonstrate the estimation makes speedup of the NLL computation.
3. Based on the formula, they propose an estimation of the matrix product of adjoint FI and certain vector only by the trained diffusion model;
    - they also analyze its estimation errors,
    - they apply the estimation to adjoint guidance sampling and show its effectiveness.
4. They provide a condition for the solution of a probability-flow ODE to become an optimal transport map.

**Strengths:**

- originality+
    - Focusing on the estimation of Fisher Information sounds novel and interesting at least for me.
- quality+
    - The paper is easy to follow, and the statements are clear, except for the theorem 1.
- significance+
    - Speedup of the NLL estimation would be useful even if it is not exact calculation. In addition, the AFI endpoint approximation seems also to be useful.

**Weaknesses:**

- clarity-
    - There were some unclear points. I will summarize them in the following questions.

**Questions:**

**Typo?**
1. In eq (1), $\alpha_t$ may be better to be replaced by $\alpha(t)$?
2. In eq (4), the symbol $s_\theta$ suddenly appeared without any explanation. I know it is related to $\epsilon_\theta$ and $\bar{y}_\theta$, but it would be better to leave some comments on them.
3. In eq (18), the symbol $h_\theta$ suddenly appeared without any explanation. I have no idea what this symbol stands for.

**Questions**
1. On JVP for trace, is it different from the Hutchinson's trace estimator?
2. On Proposition 7 and 8, the 2nd term and the 1st term of the errors seems to be related to data itself, and seems to be out of control, at least it cannot be reduced only with the improving its training. Does it cause any issue in practice?
3. In eq (20), I could not understand why the replacement $\sum_i w_i y_i y_i^\top \approx x_0 x_0^\top$ is valid.
4. In theorem 4, what distributions does this map define optimal transport between?

---

> ### Author Response · Authors · 2024-11-20
> **Reply to Reviewer jSGi (part 1)**
>
> Thank you for your appreciation of our work, we will answer your questions one by one regarding these weaknesses/questions. Due to the characters limit, we will answer in two blocks.
>
> > **Answer to Typo 1:** In eq (1), $\alpha_t$ may be better to be replaced by $\alpha(t)$?
>
> In line 94, we denote $\alpha(t)$ and $\sigma(t)$ as $\alpha_t$ and $\sigma_t$ for simplicity. And eq(1) is given before this, we agree that it may be better to replace $\alpha_t$ by $\alpha(t)$ in eq(1).
>
> > **Answer to Typo 2:** In eq (4), the symbol $s_\theta$ suddenly appeared without any explanation...
>
> We apologize for the oversight in providing necessary commentary on this. $s_\theta$ represents the parameterized score, i.e., $s_\theta(x_t,t) = -\frac{\varepsilon_\theta(x_t,t)}{\sigma(t)}$. We will include additional explanations regarding $s_\theta$.
>
> > **Answer to Typo 3:** In eq (18), the symbol $h_\theta$ suddenly appeared ...
>
> The definition of $h_\theta$ in equation (18) can be traced back to equation (6). Here, $h_\theta$ represents the neural ODE velocity representation of the PF-ODE. We apologize for any confusion caused and will provide a more comprehensive explanation near equation (18) in our revised manuscript.
>
> > **Answer to Questions 1:** On JVP for trace, is it different from the Hutchinson's trace estimator?
>
> Our JVP baseline bears similarities to Hutchinson's trace estimator, but they are not identical. Our JVP baseline method computes and sums up every element of the trace, whereas Hutchinson's method estimates the trace using a Monte Carlo approach. If we were to modify our JVP baseline to sample some indices and use their average for estimation, it would then resemble Hutchinson's method.
>
> Even with the help of the Hutchinson's method, JVP-type method can also fail in NLL tasks.  To attain a relative error less than $\epsilon$ with a probability of $1-\delta$, the Hutchinson method requires $\frac{2(1-\frac{8}{3}\epsilon)\log\frac{1}{\delta}}{\epsilon^2}$ samples [1]. Assuming that the goal is to obtain an estimation with a relative error less than 10\% with a probability exceeding 90\%, a minimum of 351 NFEs is required. This equates to 1 or 2 minutes on SD-v1.5 for a single trace estimation. For a complete NLL estimation of an image with 20 steps, this would take 20 minutes, which is entirely impractical in any business context. In contrast, our AFI-TM only requires 1 NFE for a single trace estimation, needing merely 10 seconds for the full NLL estimation of an image.
>
> In fact, there are already attempts to use the Hutchinson method to expedite the JVP of trace estimation in diffusion models [2]. Nonetheless, due to Hutchinson's limitations, these practices are restricted to relatively small DMs (CIFAR-10 at most). We will add discussions on the Hutchinson method in our revision.
>
> [1] Skorski, Maciej. "Modern analysis of hutchinson's trace estimator." 2021 55th CISS. IEEE, 2021.
>
> [2] Lu, Cheng, et al. "Maximum likelihood training for score-based diffusion odes by high order denoising score matching." International Conference on Machine Learning. PMLR, 2022.

---

> ### Author Response · Authors · 2024-11-20
> **Reply to Reviewer jSGi (part 2)**
>
> > **Answer to Questions 2:** On Proposition 7 and 8, the 2nd term and the 1st term of the errors seems to be related to data itself, and seems to be out of control, at least it cannot be reduced only with the improving its training. Does it cause any issue in practice?
>
> Yes, the current bounds in Propositions 7 and 8 do include the out-of-training control components.
>
> For Proposition 7, we derive a tighter bound that does not involve the data diameter. We find that the approximation error in Proposition 7 is at most $\frac{\alpha_t^2}{\sigma_t^4}\delta_1 +\frac{1}{\sigma_t^2}\delta_2^2$. This bound can be reduced to zero as the training loss decreases. We achieve this tighter bound by applying a more accurate triangle inequality, a change we will implement in our revision.
>
> As for Proposition 8, our current analysis only allows for the derivation of a bound that includes the data diameter. We plan to develop tighter bounds based on a more detailed analysis of the behavior of $w_i$ in the future.
>
> > **Answer to Questions 3:** I could not understand why the replacement $\sum_iy_iy_i^{\top}\approx x_0x_0^{\top}$ is valid.
>
> The replacement $\sum_iy_iy_i^{\top}\approx x_0x_0^{\top}$ originates from the observation that $w_i$ is a weighting of the summation equal to 1, and as $t$ approaches 0, the $w_i$ closest to $x_0$ will dominate due to the diffusion kernel. Therefore, this approximation is intuitively reasonable near $t=0$, which is precisely where we apply adjoint guidance.
>
> From the rigorous theoretical perspective, considering the error bounds in Propositions 7 and 8, the AFI-EA method is less valid than the AFI-TM method, as we currently cannot establish vanishing bounds for it.
>
> However, from an empirical perspective, our AFI-EA outperforms the naive JVP method in terms of score improvement across various tasks and pretrained models, as demonstrated in Figure 4. Thus, the AFI-EA approximation proves to be valid in a practical sense.
>
> > **Answer to Questions 4:** In theorem 4, what distributions does this map define optimal transport between?
>
> Are you referring to the map in Theorem 1? The map we concentrate on in Theorem 1 transitions between the marginal diffused distribution $p_t(x_t)$ at time $t$ and the marginal diffused distribution $p_s(x_s)$ at time $s$. This map's corresponding relationship is deduced from the PF-ODE in equation (6). The objective of our theorem 1 is to determine the optimal transport property of this map.

---

> ### Author Response · Authors · 2024-11-25
> **Rebuttal Revision**
>
> Dear Reviewer jSGi,
>
> We have revised our manuscript with your valuable suggestions and feedback. The revised version has been uploaded. In this updated version, we have endeavored to address all the points you raised in your review. In particular:
>
> 1. We fixed all the typos you mentioned. We sincerely apologize once again for our previous oversights.
> 2. For comparison to the Hutchinson estimator, we add more discussions in Appendix C.5.
> 3. We establish a tighter bound in Proposition 7.
> 4. We add more discussions on the non-vanishing bound of AFI-EA in Appendix C.6.
> 4. We add more discussions on the intuition of AFI-EA in section 5.
>
> All of our revised parts have been outlined in blue color. Thanks again for your helpful feedback!

---

> > ### Comment · Reviewer_jSGi · 2024-11-27
> >
> > Thank you for your replies.
> >
> > > Are you referring to the map in Theorem 1?
> >
> > Yes, sorry for it.
> >
> > I took a look on the revised version, and noticed that there is still no explanation on $s_\theta$.
> > Besides it, I think it is OK.  I think I understood the answers to my questions. Through reading it, my impression is not so largely changed, so I will keep my score.

---

> > > ### Author Response · Authors · 2024-11-27
> > >
> > > Thank you for your valuable time and your recognition of our work!
> > >
> > > We have added the explanation on $s_\theta(x_t,t)$.
> > >
> > > Best regards!

---

### Official Review · Reviewer_r2vX · 2024-11-02

**Soundness:** 2
**Presentation:** 1
**Contribution:** 2
**Rating:** 3
**Confidence:** 4

**Summary:**

This paper provides an analytical formula for the Fisher information, and uses jacobian-vector product tricks to obtain fast evaluation of likelihood evaluations. They use their formula to make a case for when, in fact, the diffusion ODE map /is/ the optimal transport map between two distributions. They have several experiments that use their approach for computing the negative loglikelihood (NLL) which appears to be correlated with high quality text-prompt/image generation.

**Strengths:**

This paper proposed a very natural solution to an otherwise computationally expensive problem.

**Weaknesses:**

The weaknesses of this paper are largely in the exposition, and in the sparsity of the literature review.

On the writing, I strongly recommend having a native English speaker make a pass through the article if possible. There are many basic syntax errors throughout the text that ought to be corrected before submission.

The formula for the Hessian of a log-convolution of a density is well-known. This is a property that simply arises because this is a log-partition function, and the covariance appears when one takes two derivatives.

I think the paper also tries to "do too much" by also trying to mention optimal transport maps --- I'm not sure what the connection is to this and other parts of the paper. I think this is also an interesting contribution to the field, but I'm not confident putting these ideas in the same work is best for clarity purposes.

**Questions:**

N/A

---

> ### Author Response · Authors · 2024-11-20
> **Reply to Reviewer r2vX**
>
> Thank you for your valuable feedback, we will answer your questions one by one regarding these weaknesses.
>
> > **Answer to Weaknesses 1:** On the writing, I strongly recommend having a native English speaker make a pass through the article if possible. There are many basic syntax errors throughout the text that ought to be corrected before submission.
>
> We admit that none of our co-authors are native English speakers and we do not have access to a researcher who is. However, we will try our best to proofread our work thoroughly and correct all syntax errors using grammar tools.
>
> > **Answer to Weaknesses 2:** The formula for the Hessian of a log-convolution of a density is well-known. This is a property that simply arises because this is a log-partition function, and the covariance appears when one takes two derivatives.
>
> Yes, our Proposition 3 is inherently the Hessian of the diffused log-density, which is a function operated by log-convolution. The mathematical technique behind it is several second derivatives. While this might not seem particularly intriguing to a theorist, within this specific context, such a derivation reveals the relationship between the diffusion Fisher information and the data distribution, thereby enabling the derivation of a dataset-based training scheme in Algorithm 1. Prior studies [1][2] on diffusion Fisher information have not delved as deeply as ours, and hence, have been unable to devise a method to directly approximate it based on the dataset. This is the key contribution of our formulations.
>
> We will include discussions to illustrate that our derivation is a special case of the formula for the Hessian of a log-convolution of a density. Any literature recommendations you could provide would be greatly appreciated.
>
> [1] Fan Bao, Chongxuan Li, Jiacheng Sun, Jun Zhu, and Bo Zhang. Estimating the optimal covariance with imperfect mean in diffusion probabilistic models. arXiv preprint arXiv:2206.07309, 2022
>
> [2] Benton, Joe, et al. "Nearly d-linear convergence bounds for diffusion models via stochastic localization." (2024).
>
>
>
> > **Answer to Weaknesses 3:** I think the paper also tries to "do too much" by also trying to mention optimal transport maps --- I'm not sure what the connection is to this and other parts of the paper. I think this is also an interesting contribution to the field, but I'm not confident putting these ideas in the same work is best for clarity purposes.
>
> We put the Theorem 1 in our paper because we found that the Fisher information has a strong connection to the optimal transport property of PF-ODE. The OT property described in Theorem 1 is a direct consequence of the Fisher information form we obtained in Proposition 1. We will add more illustration to discuss the connection of this theorem to other parts of the paper in the revision.

---

> ### Author Response · Authors · 2024-11-25
> **Rebuttal Revision**
>
> Dear Reviewer r2vX,
>
> We have revised our manuscript with your valuable suggestions and feedback. The revised version has been uploaded. In this updated version, we have endeavored to address all the points you raised in your review. In particular:
>
> 1. We fixed all the syntax errors using Grammarly, amounting to approximately 20 instances. We sincerely apologize once again for our previous oversights.
> 2. We have expanded our discussion on related works in Appendix C.4
> 3.  We also want to clarify that our primary contribution is the application of this formulation to develop effective and useful algorithms, rather than the formulation itself.
> 3. We have provided additional clarification regarding the connection of Theorem 1 to other sections in Section 6.
>
> All of our revised parts have been outlined in blue color. **We hope the above rebuttal will fully address your concerns about our work, and we would really appreciate it if you could be generous in raising the score**.

---

> ### Author Response · Authors · 2024-11-27
>
> Dear reviewer r2vX,
>
> We would like to express our gratitude once again for your time and effort in reviewing our paper and providing constructive feedback.
>
> After the rebuttal, we believe that the language and clarity of our paper have been significantly enhanced.
>
> Most importantly, we have clarified that our core contribution lies in the application of the Fisher information formulation in terms of data distribution to propose highly effective algorithms for problems that are otherwise computationally expensive.
>
> While the derivation of the Fisher information formulation in terms of data distribution may not be a novel concept in mathematics, the algorithms we have proposed based on this formulation in this study are indeed innovative.
>
> Could you kindly take the time to indicate whether we have adequately addressed your concerns, and if possible, consider raising the score? We remain open and willing to engage in further discussions.

---

### Official Review · Reviewer_QLo1 · 2024-11-04

**Soundness:** 1
**Presentation:** 3
**Contribution:** 1
**Rating:** 3
**Confidence:** 4

**Summary:**

This paper proposed a way to approximate the Jacobian of the score function in a diffusion model without taking gradients. Based on expressing the Jacobian by a conditional covariance, the paper proposed some methods to estimate the Jacobian, the trace of the Jacobian,  and matrix-vector products with Jacobian. In the end of the paper, a criterion of whether the probability flow is an optimal transport map was also claimed.

**Strengths:**

This paper is clearly written. Core ideas are presented in a very readable manner, and there are no unnecessary complications.

**Weaknesses:**

While the clarity of writing in this paper is greatly appreciated, I regret to say that this paper has some major issues that resist a more positive evaluation.

1. The key result, Proposition 3 in the paper, is well-known. Most of the theory papers on diffusion contained a version of it, e.g., Benton et al., ICLR 2024 had exactly the same result.

2. Theorem 1 is wrong, and the proof in the appendix apparently did not prove it. Instead, it was only proved that positive semi-definiteness of some unknown matrix $C(t)$ implies the flow map is an optimal transport (which is yet incorrect; should be $C(t)C(T)^{-1}$).

3. The analysis of the computational complexity of estimating trace by backpropagation did not take into account much more efficient trace estimators requiring only logarithmically many backpropagations, cf. Hutchinson estimator and its modern variants.

4. Overall, the paper lacks technical depth or insight if viewed as a theory paper, due to the reason stated above. On the other hand, it lacks comprehensive experimental validation if viewed as an empirical paper. The proposed algorithm was only evaluated on a single task, and even in this task, it can be seen from Figure 5 that the proposed algorithm has worse detail (though taking less time).

**Questions:**

I have no particular questions.

---

> ### Author Response · Authors · 2024-11-20
> **Reply to Reviewer QLo1 (part 1)**
>
> Thank you for your valuable reviews, we will answer your questions one by one regarding these weaknesses.
>
> > **Weaknesses 1:** The key result, Proposition 3 in the paper, is well-known. Most of the theory papers on diffusion contained a version of it, e.g., Benton et al., ICLR 2024 had exactly the same result. Due to the characters limit, we will answer in two blocks.
>
> Like most research on diffusion Fisher information, Lemma 5 of Benton et al., ICLR 2024 merely identifies the form of Fisher information as $-\frac{1}{\sigma_t^2}I+\frac{e^{-2t}}{\sigma_t^4}\Sigma_t$. In this case, they only specify that $\Sigma_t$ is the covariance, without providing its analytical form made from the target distribution.
>
> In contrast, our Proposition 3 goes deeper and is the first to discuss the relationship of Fisher information with respect to the data distribution. This enables the derivation of a training scheme based on the dataset. Only by obtaining the form related to the data distribution can such a training scheme be enabled. While this may seem straightforward to a mathematical theorist, this transformation can aid in developing innovative algorithms, which is our unique contribution.
>
> In fact, the diffusion model has been a highly popular field in recent years, with many different sub-fields of diffusion models showing interest in the Fisher-information of diffused distributions. These include the theory analysis sub-field you mentioned, as well as covariance estimation [1][2] and linear inverse problem sub-fields [3][4]. However, these studies did not provide the form of Fisher information made from the data distribution, and thus failed in developing the trace matching training scheme and endpoint approximation techniques like ours. We will discuss this further in the related work section in our revision.
>
> [1] Fan Bao, Chongxuan Li, Jiacheng Sun, Jun Zhu, and Bo Zhang. Estimating the optimal covariance with imperfect mean in diffusion probabilistic models. arXiv preprint arXiv:2206.07309, 2022a.
>
> [2] Fan Bao, Chongxuan Li, Jun Zhu, and Bo Zhang. Analytic-dpm: an analytic estimate of the optimal reverse variance in diffusion probabilistic models. arXiv preprint arXiv:2201.06503, 2022b.
>
> [3] Benjamin Boys, Mark Girolami, Jakiw Pidstrigach, Sebastian Reich, Alan Mosca, and O DenizAkyildiz. Tweedie moment projected diffusions for inverse problems. arXiv preprint arXiv:2310.06721, 2023.
>
> [4] Peng X, Zheng Z, Dai W, et al. Improving Diffusion Models for Inverse Problems Using Optimal Posterior Covariance[C]//Forty-first International Conference on Machine Learning. 2024.
>
> > **Weaknesses 2:** Theorem 1 is wrong, which is yet incorrect; should be $C(t)C(T)^{-1}$
>
> Your calculation is correct, but we argue that our Theorem 1 is not wrong. As shown in the Equation 70 of [1] and Equation 9 of https://math.mit.edu/~jorloff/suppnotes/suppnotes03/ls6.pdf, the definition of normalized fundamental matrix alright specify that its value at starting time should be the identity matrix $I$. That is, $C(T) = I$ (our adjoint ODE in eq(51) flows from $t=T$ to $t=s$, thus $T$ is indeed the starting time). Than your helpful calculation $C(s)C(T)^{-1} = C(s)I^{-1} = C(s)$. Your kindly provided answer is the same as ours. We apologize for not clearly state that our adjoint ODE flows from $t=T$ to $t=s$ and will make it more clear in the revision.
>
> > **Weaknesses 2:** the proof in the appendix apparently did not prove it. Instead, it was only proved that positive semi-definiteness of some unknown matrix $C(t)$ implies the flow map is an optimal transport
>
> We not only prove that our AFI-OT condition implies the flow map is an optimal transport, instead, we show that the AFI-OT condition is a sufficient as well as necessary condition. Every deduction step of our Theorem 1 is a 'if and only if' transformation. It is the first sufficient as well as necessary condition of the PF-ODE deduced map to be Monge optimal. It is very possible that, this property is indeed very complicated so that we have to use the analytical-form-unknown normalized fundamental matrix to describe it, the beautiful analytical form of $C(t)$ may not exist for a general PF-ODE at all.
>
> In fact, we did numerical experiments to approximate $C(s)$ under popular noise scheme and found it to be not p.s.d., a conclusion we are, however, unable to prove.  Therefore, we only characterize a necessary and sufficient condition for it. Future studies may leverage more advance math technique on fundamental matrix like Magnus expansion [1] to explore this topic following our path.
>
> [1] Blanes, Sergio, et al. "The Magnus expansion and some of its applications." Physics reports 470.5-6 (2009): 151-238.

---

> ### Author Response · Authors · 2024-11-20
> **Reply to Reviewer QLo1 (part 2)**
>
> > **Weaknesses 3:** The analysis of the computational complexity of estimating trace by backpropagation did not take into account much more efficient trace estimators requiring only logarithmically many backpropagations, cf. Hutchinson estimator and its modern variants.
>
> We have not discussed the Hutchinson estimator, as it is a Monte-Carlo type estimation, unlike the direct evaluation methods like full-JVP and our AFI-TM. Furthermore, the Hutchinson method and its variants are considerably more expensive than our method, and their applicability is also more restricted.
>
> - The Hutchinson method is much more costly:
>
> To attain a relative error less than $\epsilon$ with a probability of $1-\delta$, the Hutchinson method requires $\frac{2(1-\frac{8}{3}\epsilon)\log\frac{1}{\delta}}{\epsilon^2}$ samples [1].
> Assuming that the goal is to obtain an estimation with a relative error less than 10\% with a probability exceeding 90\%, a minimum of 351 NFEs is required. This equates to 1 or 2 minutes on SD-v1.5 for a single trace estimation. For a complete NLL estimation of an image with 20 steps, this would take 20 minutes, which is entirely impractical in any business context. In contrast, our AFI-TM only requires 1 NFE for a single trace estimation, needing merely 10 seconds for the full NLL estimation of an image.
>
> - The application of the Hutchinson method is more restricted:
>
> Due to its Monte-Carlo characteristics, the Hutchinson method is more appropriate for contributing to the computation of certain training objectives as in [2], where the unbiased property is sufficient, and large variance may be absorbed into network training. However, the Hutchinson method may encounter difficulties with accurate per-sample trace estimation. Our method can accommodate both scenarios, including per-sample computation.
>
>
> In fact, there are already attempts to use the Hutchinson method to expedite the JVP of trace estimation in diffusion models [2]. Nonetheless, due to Hutchinson's limitations, these practices are restricted to relatively small DMs (CIFAR-10 at most). We will  add discussions on the Hutchinson method in our revision.
>
> [1] Skorski, Maciej. "Modern analysis of hutchinson's trace estimator." 2021 55th CISS. IEEE, 2021.
>
> [2] Lu, Cheng, et al. "Maximum likelihood training for score-based diffusion odes by high order denoising score matching." International Conference on Machine Learning. PMLR, 2022.
>
> > **Weaknesses 4:** it lacks comprehensive experimental validation if viewed as an empirical paper. The proposed algorithm was only evaluated on a single task
>
> While we acknowledge that further experiments could be beneficial, but please allow me to humbly disagree that our empirical results are limited.
> We do skip the experiments on academic-level diffusion models. Instead, we tested our proposed methods, both AFI-TM and AFI-EA, on the most advanced commercial-level diffusion models with open accessible training dataset (more advanced commercial diffusion models, like FLUX, SD-XL or SD-v3, do not open-source their training dataset).
> Our experiments involve training details, quantitative NLL evaluation results, and a qualitative test on the trade-off curve of NLL and CLIP for our AFI-TM method. We also tested four different sub-tasks for our AFI-EA method in the adjoint guidance experiments, which aligns with the standard practices of adjoint guidance papers [1][2][3].
> However, we appraciate the advice on more experiments and will conduct additional tests on other diffusion models.
>
> [1] Jiachun Pan, Jun Hao Liew, Vincent YF Tan, Jiashi Feng, and Hanshu Yan. Adjointdpm: Adjoint sensitivity method for gradient backpropagation of diffusion probabilistic models. arXiv preprint arXiv:2307.10711, 2023
>
> [2] Jiachun Pan, Hanshu Yan, Jun Hao Liew, Jiashi Feng, and Vincent YF Tan. Towards accurate guided diffusion sampling through symplectic adjoint method. arXiv preprint arXiv:2312.12030, 2023.
>
> [3] Zander W Blasingame and Chen Liu. Adjointdeis: Efficient gradients for diffusion models. arXiv preprint arXiv:2405.15020, 2024
>
> > **Weaknesses 4:**  even in this task, it can be seen from Figure 5 that the proposed algorithm has worse detail (though taking less time).
>
> Figure 5 demonstrates that our AFI-EA method can enhance the aesthetic score of the generated images while reducing time costs. The smoother visual effect observed in the resulting images is a desirable outcome of the target aesthetic score estimator [1], but it is not a direct result of our AFI-EA method. Our method ensures the production of better images according to the criteria of the selected score functions, in less time.
>
> [1] https://github.com/christophschuhmann/improved-aesthetic-predictor

---

> ### Comment · Reviewer_QLo1 · 2024-11-20
>
> I disagree that Proposition 3 contains anything that was not in Lemma 5 of Benton et al., ICLR 2024. The possibility of deriving a training scheme is obvious from that lemma, just in the same way that the possibility of learning score function is obvious from Tweedie's formula.
>
> I insist that Theorem 1 is wrong (or useless, depending how you define "fundamental solutions"). But to make the discussion more constructive, could you first write down your definition of fundamental solutions? In doing so, please be aware that $B(t)$ depends also on $x$. The theorem becomes useless once the correct definition is used, as it is merely saying that the flow map is an OT if it has p.s.d. Jacobian, which is a standard fact, and that the Jacobian is the fundamental solution of an ODE, which is trivial (this is known to hold for every ODE, not just probability flow).

---

> ### Author Response · Authors · 2024-11-22
> **Reply to Reviewer QLo1 (Continual, part1)**
>
> Thanks for your reply! Your feedback is valuable. Now, we would like to address the concerns you raised in two blocks.
>
> > 1. I disagree that Proposition 3 contains anything that was not in Lemma 5 of Benton et al., ICLR 2024.
>
> Proposition 3 presents the specific form of Fisher information in terms of **data distribution**, which is not included in Lemma 5 of Benton et al., ICLR 2024. This distinction is crucial as it facilitates the derivation of our new algorithms.
> Also, we adopt the currently more commonly used $\alpha_t,\sigma_t$ notation to represent the noise schedule. Instead, Benton et al., ICLR 2024 fix $\alpha_t\equiv e^{-2t}$.
> We gave out this detailed formulation for avoiding unnecessary misunderstandings in the development of the subsequent training scheme. As a counter-example, we reach to misunderstandings on theorem 1 because we didn't gave the detailed formulation of the normalized fundamental matrix.
>
>
> Anyway, we will declare that our Proposition 3 is a slightly transformed version of Lemma 5 of Benton et al., ICLR 2024. This is not our core contribution. Our primary contribution lies in leveraging this formulation to derive two efficient algorithms.
> And we have to present Proposition 3 in our paper for completeness, as it serves as the foundation for deriving our novel algorithms, AFI-TM and AFI-EA.
> We kindly request a fair evaluation based on the overall content and contributions of our paper.
>
> > 2. The possibility of deriving a training scheme is obvious from that lemma, just in the same way that the possibility of learning score function is obvious from Tweedie's formula.
>
> Yes, our algorithm is straight-forward and can be easily understand, which leverage the analytical form of the Fisher information to develop training scheme for trace matching and consequently efficient NLL evaluation.  Straight-forward is not a bad thing.
>
> Our novelty lies in the effectiveness of the proposed AFI-TM and AFI-EA algorithms, which has not been explored previously, neither by [Benton et al., ICLR 2024], nor by other researchers.
>
> In fact, there are still very recent efforts, such as [1], striving to propose methods capable of evaluating NLL alongside the generated samples of diffusion models, even if it requires training entirely new diffusion models.
>
> [1] Yadin S, Elata N, Michaeli T. Classification Diffusion Models: Revitalizing Density Ratio Estimation[C] NeurIPS 2024.

---

> ### Author Response · Authors · 2024-11-22
> **Reply to Reviewer QLo1 (Continual, part2)**
>
> > 3. I insist that Theorem 1 is wrong (or useless, depending how you define "fundamental solutions"). But to make the discussion more constructive, could you first write down your definition of fundamental solutions? In doing so, please be aware that $B(t)$ depends also on $x$. The theorem becomes useless once the correct definition is used, as it is merely saying that the flow map is an OT if it has p.s.d. Jacobian, which is a standard fact, and that the Jacobian is the fundamental solution of an ODE, which is trivial (this is known to hold for every ODE, not just probability flow).
>
> My apologies if our theorem proof and definitions were not clear.
>
> First, we understood that a flow map is an OT if it possesses a p.s.d. Jacobian. We denote this Jacobian as $A(t) = \frac{\partial T_{t,T}(x_t)}{\partial x_t}$.
> Our objective is to discuss the conditions under which $A(t)$ can be p.s.d.
> We then employ the adjoint technique, as illustrated in eq(51), to ascertain that $A(t)$ is, in fact, the solution to the following initial value problem
> $$
> (\text{ODE 1}),\quad \frac{d A(t)}{d t} = A(t)B(t), A(T) = I,\quad(\text{flow from T to t})
> $$
> Let's define $C(t)$ as the normalized fundamental matrix at $t$ for $B(t)$. This implies $C(t)$ is the solution to the following ODE:
> $$
> (\text{ODE 2}),\quad C'(t) = B(t)C(t),C(T)=I,\quad(\text{flow from T to t})
> $$
> According to eq(10) in [1], in this scenario, the solution to (ODE 1) is given by: $A(t) = C(t)A(T) = C(t)$. As a result, the flow map is an OT iff $A(t)$ is p.s.d. iff $C(t)$ is p.s.d.
>
> Unfortunately, we cannot obtain the analytical form of either $A(t)$ or $C(t)$. However, one can either attempt to numerically solve for $A(t)$ using our (ODE 1), or delve deeper into $C(t)$ by leveraging techniques associated with the normalized fundamental matrix on our (ODE 2), such as the Magnus expansion [2].
>
> We acknowledge that many of our readers may not be familiar with the concepts of normalized fundamental matrix or adjoint ODE. Therefore, we will provide a more comprehensive illustration of these concepts in our revision.
>
> Notice that, the above requirement need to be satisfied for every $x$. We apologize for the oversight in providing necessary commentary on this in the original paper. We will make it more clear in revision.
>
> Based on Theorem 1, we conducted  numerical experiments (low-dimensional several-data-level) and discovered that for the PF-ODE, if the training data does not fall on a straight line, the matrix is indeed significantly non-symmetric. The error exceeds the bound of the ODE numerical error.
> Therefore, our theorem can enable the usage of numerical experiments to verify whether the PF-ODE is an OT given the data.
> This is meaningful for OT-related generative model area as well.
>
>
> [1] https://math.mit.edu/~jorloff/suppnotes/suppnotes03/ls6.pdf
>
> [2] Blanes, Sergio, et al. "The Magnus expansion and some of its applications." Physics reports 470.5-6 (2009): 151-238.

---

> > ### Comment · Reviewer_QLo1 · 2024-11-22
> >
> > Your clarification is greatly appreciated. The derivation is still incorrect, as the ODE should be $\frac{dA(t, x)}{dt} = A(t, x) B(t, T_{t, T}(x))$. Anyway, I am not convinced that Theorem 1 is useful. It is impossible to check the criterion for every $x$, and in every scenario where the criterion is practically relevant, it is much easier to compute the Jacobian of the flow map directly.

---

> ### Author Response · Authors · 2024-11-22
> **Reply to Reviewer QLo1 (Continual 2)**
>
> Thanks for your fast response! We would like to address the concerns.
>
> > 1.  The derivation is still incorrect, as the ODE should be $\frac{dA(t, x)}{dt} = A(t, x) B(t, T_{t, T}(x))$.
>
>
> We made a second examination, we would like to clarify that the $x$ won't move in the ODE 1.
> Recall our ODE 1:
> $$
> (\text{ODE 1}),\quad \frac{d A(t)}{d t} = A(t)B(t), A(T) = I,\quad(\text{flow from T to t})
> $$
> The ODE 1 is a result of the following derivation:
> $$
> \begin{aligned}
> \frac{\mathrm{d} {A}(t)}{\mathrm{d} t} & =\lim _{\epsilon \rightarrow 0^{+}} \frac{{A}(t+\epsilon)-{A}(t)}{\epsilon} \\
> \\\\
> &=\lim _{\epsilon \rightarrow 0^{+}} \frac{{A}(t+\epsilon)-{A}(t+\epsilon)\nabla_x[ {x}+\epsilon(f(x)-g^2(t)/2\nabla_x\log q_t(x)) +\mathcal{O} (\epsilon^2  )] }{\epsilon}
> \\\\
> &=\lim _{\epsilon \rightarrow 0^{+}}\frac{\epsilon f(t)A(t+\epsilon)+g^2(t)/2\epsilon A(t+\epsilon)\nabla_x\log q_t(x)+\mathcal(O)(\epsilon^2)}{\epsilon}
> \\\\
> &=f(t)A(t)+\frac{g^2(t)}{2}A(t)\nabla_x \log q_t(x)
> \\\\
> &=A(t)B(t)
> \end{aligned}
> $$
> All the steps here are all infinitesimal analysis near $x$, there is no actual moving of $x$. This is the reason why we leave out the $x$ notation in $A(t)$, $B(t)$ and $C(t)$ for simplicity. Thus for any $x\in\mathbb{R}^n$, we have $\frac{dA(t, x)}{dt} = A(t, x) B(t, x)$.
> Sorry for the confusion we make, we will add more illustration in the revision.
>
> > 2. Anyway, I am not convinced that Theorem 1 is useful. It is impossible to check the criterion for every $x$
> , and in every scenario where the criterion is practically relevant, it is much easier to compute the Jacobian of the flow map directly.
>
> We cannot check this criterion for every $x$ to prove that the PF-ODE deduced map is an OT.
> However, we can check only one $x$, that this condition does not hold on this $x$, to prove that the PF-ODE deduced map is absolutely not an OT.
> This is the usefulness of our Theorem 1, we actually prove an equivalence condition for the OT property of the PF-ODE deduced map. We are not just give a sufficient condition for the OT property of the PF-ODE deduced map.
>
> Personally, I don't really think that a PF-ODE deduced map can really be an OT in the non-trivial cases. Because I did some numerical experiments on very simple 2D dataset (even only five data points), and this condition still fail, the $A(t)$ is shown to be notably non-symmetric. Maybe only the very trivial case mentioned in [1] can make the PF-ODE deduced map to be an OT.
> Of course, this part is only an informal discussion.
>
> [1] Zhang, Pengze, et al. "Formulating discrete probability flow through optimal transport." Advances in Neural Information Processing Systems 36 (2024).

---

> > ### Author Response · Authors · 2024-11-27
> >
> > Dear reviewer QLo1,
> >
> > We would like to thank you again for your time in reviewing our paper and providing constructive criticism. After the discussion, we believe that the clarification of our theorem and illustration of our core contribution is expressed. Would you have the time to indicate whether we have addressed your concerns (to raise the score)? We are also happy to discuss more.

---

> > > ### Comment · Reviewer_QLo1 · 2024-11-27
> > >
> > > Thank you for your response. However, I believe Theorem 1 is incorrect and the ODE should be something like $\frac{dA(t, x)}{dt} = A(t, x) B(t, T_{t, T}(x))$ (there might be some extra factors). This follows from the elementary fact that the flow map $T_{s, t}$ of the ODE $dx_t = v(t, x_t) dt$ satisfies $\frac{\partial T_{s, t}(x)}{\partial t} = v(t, T_{s, t}(x))$.

---

> ### Author Response · Authors · 2024-11-25
> **Rebuttal Revision**
>
> Dear Reviewer QLo1,
>
> We have revised our manuscript with your valuable suggestions and feedback. The revised version has been uploaded. In this updated version, we have endeavored to address all the points you raised in your review. In particular:
>
> 1.  We state that several related works are proposing a similar formulation to our Proposition 3, and our formulation can be obtained by transformation on theirs in section 3 and Appendix C.4.
> 2.  For Theorem 1, we add more illustrations and definitions in Appendix A.10.
> 3. For comparison to the Hutchinson estimator, we add more discussion in  Appendix C.5.
> 4. For more experimental details and clarification on the visual effect of Figure 5, we add more discussion in the caption of Figure 5 and Appendix B.2.
>
> All of our revised parts have been outlined in blue color. Thank you for your effort in our discussion! **We hope the above discussion will fully address your concerns about our work, and we would really appreciate it if you could be generous in raising the score.**

---

> ### Author Response · Authors · 2024-11-27
>
> Thanks for your response!
>
> Your formulation is calculating the partial derivative of Jacobian w.r.t. to $t$.
>
> However, our adjoint ODE first fixes a $x$ and then calculates the change of Jacobian w.r.t. $t$.
>
> They are different.
>
> You can trust our adjoint derivation. This technique is commonly used. Our derivation is borrowed from the eq(39-45) of [1]. [1] is a very renowned paper within the neural ODE field. Their eq(39-45) do not involve state $z(t)$ at other times, only $z(t)$ at current time is needed.
>
> [1] Chen, Ricky TQ, et al. "Neural ordinary differential equations." NeurIPS 31 (2018).

---

> > ### Comment · Reviewer_QLo1 · 2024-11-27
> >
> > Thank you for your clarification. Your proof seems to be messing around with notations. The equation $\frac{dA(t, x)}{dt} = A(t, x) B(t, T_{t, T}(x))$ can be readily obtained with a one-line computation by differentiating $\frac{\partial T_{s, t}(x)}{\partial t} = v(t, T_{s, t}(x))$.  The reference you provided, unfortunately, confirms that your derivation is wrong. Apparently, (39-45) in [1] are for backpropagation, where the gradient is computed for $z(t)$, which is not relevant to computing $\partial T_{t, T}(x)/\partial x$ where the variable $x$ has no dependence on $t$.
> >
> > It is easy to check that Theorem 1 is wrong on a balanced 2-GMM, which I suggest the authors add as an example to the paper.

---

> ### Author Response · Authors · 2024-11-28
>
> Thanks for your reply and valuable discussion.
>
> Sorry that parts of our previous clarification are not rigorous.
>
> Our correct definition should be: $A(t) = \frac{\partial T_{t,T}(x_t)}{\partial x_t}$ and $B(t) = B(t,x_t)$, they should be defined relating to $x_t$.
>
> In this case, our derivation is in alignment with eq(39-45) in [1].
> Our adjoint measures:
> $$
> \frac{dA(t)}{dt} =  \frac{\partial T_{t,T}(x_t)}{\partial x_t}/d t
> $$
> and eq(39-45) in [1] measures:
> $$
> \frac{d a(t)}{dt} = \frac{\partial L(z_t)}{\partial z_t}/d t
> $$
> our ODE is a matrix extension of theirs.
>
> Our adjoint ODE is performed based on a whole sampling path $x_t, t \in [T,s]$, not a fix $x$.
> Theorem 1 should discuss all PF-ODE chains starting from a $x_T\in\mathbb{R}^d$
> In this way, our Theorem is correct, this is exactly what we presented at first. But in the discussion, I misunderstood what my coauthor meant and thought it was based on a fixed $x$.
>
> We have now made this justification in our revised paper.
>
> Thank you for your valuable advice again, now whether we have addressed your concerns?
>
> [1] Chen, Ricky TQ, et al. "Neural ordinary differential equations." NeurIPS 31 (2018).

---

> > ### Comment · Reviewer_QLo1 · 2024-11-28
> >
> > I regret to say that as I have mentioned, this is just messing around with notations and is doing nothing more than the well-understood equation $\frac{dA(t, x)}{dt} = A(t, x) B(t, T_{t, T}(x))$, which is useless as I have argued.

---

> ### Author Response · Authors · 2024-11-28
>
> Thanks for your reply.
>
> We wish to clarify that we define the $A(t) = \frac{\partial T_{t,T}(x_t)}{\partial x_t}$, there is no $A(t,x)$, $A(\cdot)$ is only relating to $t$, as an arbitrary initial point $x_T$ of PF-ODE is selected (or equivalently, a PF-ODE chain is selected).
>
> In our notation, there is only this form:
> $$
> \frac{dA(t)}{dt} = A(t) B(t) \text{ (derived by our eq(51))}
> $$
> where $B(t) = B(t,x_t)$. Please refer to our eq(22), the $B(t,x_t)$ depends on $x_t$ because of containing the term $w_i(x_t)$, and thus relates to $x_t$, and finally can only depend on $t$, as long as a PF-ODE chain is selected.
>
> $B(t)$ has nothing to do with $T_{t,T}(\cdot)$.
>
> Please note that, as an arbitrary initial point $x_T$ of PF-ODE is selected, $x_t$ can totally depend on $t$.

---

> > ### Comment · Reviewer_QLo1 · 2024-11-28
> >
> > I will stop responding as the discussion is becoming increasingly unfruitful. The authors did not attempt to figure out the obvious one-line computation or to work out the 2-GMM example, and instead kept messing around with notations. $B(t, x_t)$ is simply $B(t, T_{t, T}(x))$ as $x_t = T_{t, T}(x)$, and it serves no purpose to keep using this $x_t$ notation which is likely the source of all these errors.

---

> ### Author Response · Authors · 2024-11-28
>
> Dear Reviewer QLo1,
>
>  I appreciate our academic discussion and understand the importance of maintaining a constructive tone, there is no need for any of us to lose our temper. Wish you a good day.
>
> While I comprehend your request for elaboration on the 'one-line computation' and '2-GMM', I must admit that I'm experiencing some confusion.
>
> > The equation $\frac{dA(t, x)}{dt} = A(t, x) B(t, T_{t, T}(x))$ can be readily obtained with a one-line computation by differentiating $\frac{\partial T_{s, t}(x)}{\partial t} = v(t, T_{s, t}(x))$.
>
> I understand that $x_t = T_{T, t}(x_T)$. Maybe you mean there is a more convenient way to derive our eq(51). Do you indicate that we can get our eq(51) by applying partial differentiation to $x_t$ on both sides of $\frac{\partial T_{s, t}(x)}{\partial t} = v(t, T_{s, t}(x))$?  But I am not sure what is the specific form of $v$ and how to finish this one-line computation to get the form of $A(t)$.
>
> Our proof in eq(51)  deduced $\frac{dA(t)}{dt} = A(t) B(t)$ based on the adjoint derivation borrowed from [1], please let us know whether you agree that this derivation is correct or not, based on the notation defined in theorem 1.  We want to emphasize again that our $B(t) = B(t,x_t)$ is determined by the time-related $x_t$ because its analytical form shown in eq(22) involves $w_i(x_t)$.
>
> [1] Chen, Ricky TQ, et al. "Neural ordinary differential equations." NeurIPS 31 (2018).
>
>
> > It is easy to check that Theorem 1 is wrong on a balanced 2-GMM, which I suggest the authors add as an example to the paper.
>
> I am not sure what you mean by 'is wrong'. Are you suggesting that we should test when the data distribution is in 2-GMM, and subsequently discover that the condition in Theorem 1 is not met? If so, this would only indicate that the PF-ODE mapping is not OT, rather than implying that our Theorem 1 is incorrect. Our Theorem simply proposes an 'if and only if' relationship between proposition A and proposition B. We have not asserted that either proposition A or B must necessarily be true or false in any given scenario.
>
> Anyway, usefulness is a rather subjective term, to make our discussion more fruitful, let's focus on the correctness first. I am confident that the derivation and notations used in our paper are consistent and correct. Could you please confirm whether you agree with the correctness of our derivation, or point out any steps in the following you believe to be incorrect?
>
>
> - Step 0: We fix a specific PF-ODE chain starting from some $x_T$.
> - Step 1: We define $A(t)$ to be $\frac{\partial T_{t,T}(x_t)}{\partial x_t}$ and the deduced map is OT iff $A(t)$ is p.s.d. (Why not $A(t,x_t)$? Because the $x_t$ is specified by the PF-ODE together with time $t$, so we can write it simply as $A(t)$)
> - Step 2: We model the change of $A(t)$ from $T$ to $t$, from $x_T$ to $x_t$ along the PF-ODE chain, as an IVP. The initial value, $A(T)$, is of course $I$. The ODE dynamic of this IVP is deduced in eq(51). That is, $\frac{d A(t)}{d t} = A(t)B(t)$, where $B(t)$ can depend only on $t$ as the PF-ODE chain is fixed.
> - Step 3: Based on the IVP on $A(t)$, and the definition of normalized fundamental matrix and eq(53), we deduce our AFI-OT condition, which is an equivalence condition of OT property of PF-ODE map.

---

> ### Author Response · Authors · 2024-12-02
> **Disagreement summarization (Part 1)**
>
> **********
> Dear Reviewers QLo1, other Reviewers, ACs, SACs and PCs,
>
> Our discussions with reviewer QLo1 have resulted in a number of unresolved disagreements, which we will summarize and clarify here.
> We will address each of the three disagreements in turn.
> Other reviewers, ACs, SACs, and PCs may make a verdict based on this summarization.
> Should the reviewer QLo1 have additional points of contention, they may add them using the following format to streamline the discussion:
>
> Disagreement  1: $B(t)$ should explicitly depend on $x_t$, meaning $B(t)$ should be represented as $B(x_t, t)$.
>
> Disagreement  2: The derivation appears to be messing. The relation $\frac{d A(x_t, t)}{d_t} = A(x_t, t) B(x_t, t)$ can be derived with a one-line computation.
>
> Disagreement  3: The utility of Theorem 1 is questionable, given the presence of $x_t$ in $B(t, x_t)$.
>
> **Disagreement 1**
>
> We acknowledge that it should be $B(x_t, t)$. As per the definition of $w_i$ in Proposition 1, $B(x_t, t)$ is dependent on $x_t$ through $w_i(x_t, t)$. Additionally, $C(x_t, t)$ should be symmetric, not merely semi-positive definite. We will now prove Theorem 1 with $B(t)$ replaced by $B(x_t, t)$.
>
> *proof*
>
> Consider the Probability Flow ODE(PF-ODE)
>
> $$
> \begin{equation}
> \frac{d x_t}{dt} = v(x_t, t).\quad\quad\quad\quad (1)
> \end{equation}
> $$
>
> This is defined over the interval $[\tau, T]$. For any initial condition $x_\tau = x \in \mathbb{R}^d$, we aim to compute $\frac{\partial x_T}{\partial x}$ and prove it is symmetric and semi-pos-def.
>
>
> Step 1.
>
> Define $x_t(x, \tau) = x + \int_{\tau}^{t} v(x_s, s) ds$ as the solution to equation (1). Then, $x_T(x, \tau) = x_T(x_t, t)$ for all $t \in [\tau, T]$. Moreover, due to the uniqueness of the solution, $x_T(x_u(x, \tau), u) = x_T(x, \tau)$ for all $u \in [\tau, T]$.
>
>
> Step 2.
>
> The term $\frac{\partial x_T(x_t, t)}{\partial x_t}$ is defined on the solution curve $x_t(x, \tau)$ and can therefore be considered as a function dependent only on $t$, given the initial condition $x_\tau = x$. The boundary condition is easily found to be $ \frac{\partial x_T(x_t, t)}{\partial x_t} \rvert_{t=T} = I $.  As this is a function of $t$, its derivative can be calculated using the chain rule $\frac{\partial x_T(x_t, t)}{\partial x_t} = \frac{\partial x_T(x_{t+\epsilon}, t + \epsilon)}{\partial x_{t + \epsilon}} \frac{\partial x_{t+\epsilon}(x_t, t)}{\partial x_{t}}$:
>
> $$
> \begin{equation}
> \begin{aligned}
> \frac{d }{dt} \frac{\partial x_T(x_t, t)}{\partial x_t} &= \lim_{\epsilon \rightarrow 0} \frac{\frac{\partial x_T(x_{t+\epsilon}, t+\epsilon)}{\partial x_{t + \epsilon}} - \frac{\partial x_T(x_t, t)}{\partial x_t}}{\epsilon} \\\\
> &= \lim_{\epsilon \rightarrow 0} \frac{1}{\epsilon} \frac{\partial x_T(x_{t+\epsilon}, t + \epsilon)}{\partial x_{t + \epsilon}} (\frac{\partial x_{t+\epsilon}(x_t, t)}{\partial x_{t}} - I) \quad\quad\quad\quad (2)\\\\
> &= \lim_{\epsilon \rightarrow 0} \frac{1}{\epsilon} \frac{\partial x_T(x_{t+\epsilon}, t + \epsilon)}{\partial x_{t + \epsilon}} (I + \int_{t}^{t+\epsilon} \frac{\partial v(x_s, s)}{\partial x_t} ds - I) \\\\
> &= \frac{\partial x_T(x_{t}, t)}{\partial x_{t}} \frac{\partial v(x_t, t)}{\partial x_t}.
> \end{aligned}
> \end{equation}
> $$
> The term $B(x_t, t)$ in Theorem 1 is given by $\frac{\partial v(x_t, t)}{\partial x_t}$. This leads us to the ODEs
>
> $$
> \begin{equation}
> \begin{cases}
> & \frac{d x_t}{dt} = v(x_t, t), \\
> &\frac{d A(x_t, t)}{dt} = A(x_t, t) B(x_t, t), \quad\quad\quad\quad (3)
> \end{cases}
> \end{equation}
> $$
> with the boundary conditions $x_\tau = x$ and $A(x_T, T) = I$.
>
> Step 3.
>
> Let $C(x_t, t)$ be the solution of equation (3). Applying Theorems 2, 3, and 4, we arrive at the final result.

---

> ### Author Response · Authors · 2024-12-02
> **Disagreement summarization (Part 2)**
>
> **Disagreement 2**
>
> There are two approaches to proving Theorem 1. We have chosen to use the backward method, although a forward method is also available. Given that $x_t(x, \tau) = x + \int_{\tau}^t v(x_s, s) ds$, we can derive that $\frac{\partial x_t}{\partial x} = I + \int_{\tau}^t \frac{\partial v(x_s, s)}{\partial x_s} \frac{\partial x_s}{\partial x} ds$ and $\frac{d}{dt} \frac{\partial x_t}{\partial x} = \frac{\partial v(x_t, t)}{\partial x_t} \frac{\partial x_t}{\partial x}$. This also leads to the same result.
>
> We opted for the backward method because it aligns well with the well-known Neural ODE[1] in the field of machine learning. In fact, our derivation essentially replaces the scalar-valued function $L$ with a vector-valued function $x_T$ in Equation (3) of [1]. (In the Neural ODE, they use $x_1$ instead of $x_T$).
>
> **Disagreement 3**
>
> With the expression for $B(x_t, t)$, we can determine the conditions under which the PF-ODE is OT or not. Here are two examples.
>
> Example (1) Given that
>
> $$
> \begin{equation}
> \begin{aligned}
> &\sum_i w_i(x_t, t) y_i y_i^T - (\sum_i w_i(x_t, t) y_i)(\sum_j w_j(x_t, t) y_j)^T \\\\
> &= \frac{1}{2} (\sum_i w_i(x_t, t) y_i y_i^T + \sum_j w_j(x_t, t) y_j y_j^T) - (\sum_i w_i(x_t, t) y_i)(\sum_j w_j(x_t, t) y_j)^T \quad\quad\quad\quad (4) \\\\
> &= \frac{1}{2} \sum_{i,j} w_i(x_t, t) w_j(x_t, t) (y_i - y_j) (y_i - y_j)^T,
> \end{aligned}
> \end{equation}
> $$
>
> we can deduce that
>
> $$
> \begin{equation}
> B(x_t, t) = [f_t - \frac{g^2_t}{2\sigma_t^2}] I + \frac{\alpha_t^2 g_t^2}{4\sigma_t^4} \sum_{i,j} w_i(x_t, t) w_j(x_t, t) (y_i - y_j) (y_i - y_j)^T. \quad\quad\quad\quad (5)
> \end{equation}
> $$
>
> In this case, $x_t$ only appears in the coefficients of the matrices, not directly in the matrix elements. This is a significant property that does not appear in general ODEs. If all the training data lie on a line, then the matrix $(y_i - y_j) (y_i - y_j)^T$ commutes with $(y_k - y_l) (y_k - y_l)^T$ for all $i,j,k,l$. Consequently,
>
> $$
> \begin{equation}
> \begin{aligned}
> &B(x_t, t) B(x_s, s) \\\\
> &= ([f_t - \frac{g^2_t}{2\sigma_t^2}] I + \frac{\alpha_t^2 g_t^2}{4\sigma_t^4} \sum_{i,j} w_i(x_t, t) w_j(x_t, t) (y_i - y_j) (y_i - y_j)^T) ([f_s - \frac{g^2_s}{2\sigma_s^2}] I + \frac{\alpha_s^2 g_s^2}{4\sigma_s^4} \sum_{i,j} w_i(x_s, s) w_j(x_s, s) (y_i - y_j) (y_i - y_j)^T)  \quad\quad\quad\quad (6) \\\\
> & = B(x_s, s) B(x_t, t).
> \end{aligned}
> \end{equation}
> $$
>
> According to Theorem 4 of [2], $C(x_\tau, \tau) = e^{\int_{\tau}^T B(x_s, s) ds}$ is symmetric and semi-positive definite, leading to an OT result. Furthermore, the fact that $(y_i - y_j) (y_i - y_j)^T$ commutes with $(y_k - y_l) (y_k - y_l)^T$ is also the sufficient condition for all training data to lie on a line. This is the only situation where an OT plan can be identified by examining the commutation of each $(y_i - y_j) (y_i - y_j)^T$ in $B$.
>
> Example (2). The coefficient $\frac{\alpha_t^2 g_t^2}{4\sigma_t^4} = -\frac{2 \alpha_t \alpha_t'}{4\sigma_t^4}$ tends to zero as $t \rightarrow 1$. Additionally, $w_i(x_t, t)$ approaches a uniform distribution with respect to $i$ as $t \rightarrow 1$. These two factors cause $B(x_s, s)B(x_t, t) - B(x_t, t) B(x_s, s)$ to approach zero, leading to an approximate OT result. This provides an explanation for the numerical experiments in [3], which show that the PF-ODE is nearly an OT plan.
>
>
> [1] Chen, Ricky TQ, et al. "Neural ordinary differential equations." NeurIPS 31 (2018).
>
> [2] Blanes, Sergio, et al. "The Magnus expansion and some of its applications." Physics reports 470.5-6 (2009): 151-238.
>
> [3] Khrulkov V, Ryzhakov G, Chertkov A, et al. Understanding DDPM Latent Codes Through Optimal Transport. ICLR 11 (2023).

---

> > ### Comment · Reviewer_QLo1 · 2024-12-02
> >
> > The authors' effort to address my concerns are greatly appreciated. I'm happy to see that Theorem 1 is corrected now, so I am willing to raise my score.
> >
> > My other concerns remain. The significance of Theorem 1 is limited, and the claim "$x_t$ only appears in the coefficients of the matrices, not directly in the matrix elements, which is a significant property" does not sound justified: it is actually a sum of a few matrices that has ``coefficients'' depending on $x_t$, which is not special since every matrix can be written in this way by decomposing it into the sum of its individual entries.

---

> ### Author Response · Authors · 2024-12-02
>
> Dear Reviewer QLo1,
>
> Thanks for your kind reply! We are happy to address your concerns on the correctness of Theorem 1.
> We are willing to engage in further clarification of your other concerns.
>
> > ... the claim "$x_t$ only appears in the coefficients of the matrices, not directly in the matrix elements, which is a significant property" does not sound justified: it is actually a sum of a few matrices that has ``coefficients'' depending on $x_t$
> , which is not special since every matrix can be written in this way by decomposing it into the sum of its individual entries.
>
> From the linear algebra view, you mentioned "every matrix can be written in this way by decomposing it into the sum of its individual entries", this is the standard basis (also called natural basis or canonical basis) decomposition on the vector space  $\mathbb{R}^{d\times d}$. This can only indicate that the matrix has a rank between 0 to $d\times d$.
>
> However, in our decomposition, every matrix is not an ordinary matrix, but a symmetric matrix of rank 1 with the form of $\\{(y_i-y_j)(y_i-y_j)^\top\\}$. It is precisely because the rank is 1 and symmetric that it is possible to deduce that PF ODE is OT when all the data is on a line. This is a distinct property from the general ODE.
> Two examples from the last reply show that theorem 1 helps to understand PF ODE from an OT point of view. It is also an addition to the current paper on the relationship between diffusion models (including flow matching and rectified flow) and OT [1][2].
>
> Moreover, implies by our Theorem 1, if the dataset scale $n$ satisfies $n^2<d^2$, we would know that every matrix along the solution path is all spaned by a lower-rank base made of $\\{(y_i-y_j)(y_i-y_j)^\top\\}_{i,j=1...n}$.
> This special property is not possessed by every matrix.
> Our theorem uncovers the relation between the rank of initial data to the rank of diffused data.
> This low-rank property is interesting for representation learning research in diffusion models such as [3].
>
> This low-rank sub-manifold understanding may explain why diffusion models can overcome the curse of dimensionality and learn the pattern of high-dimensional distributions.
> This may also justify some low-rank assumptions of diffusion model research.
> This low-rank property, which does not appear in general ODEs, also confirms the significance of our Theorem.
>
> [1] Liu, Qiang. "Rectified flow: A marginal preserving approach to optimal transport." arXiv preprint arXiv:2209.14577 (2022).
>
> [2] Khrulkov, Valentin, et al. "Understanding ddpm latent codes through optimal transport." arXiv preprint arXiv:2202.07477 (2022).
>
> [3] Wang P, Zhang H, Zhang Z, et al. Diffusion models learn low-dimensional distributions via subspace clustering[J]. arXiv preprint arXiv:2409.02426, 2024.
>
> >  The significance of Theorem 1 is limited.
>
> Please allow me to humbly argue that it is quite unfair to reject the whole paper because one-third of our contributions are thought to be 'correct but less significant'. Our proposed algorithms in this paper, AFI-TM, and AFI-EA, are appreciated as novel and efficient alternatives to otherwise computationally expensive problems by Reviewer [r2vX], Reviewer [jSGi], and Reviewer [iHvF]. We firmly believe that the value of these contributions should not be underestimated.
> Given these points, we earnestly implore you to reconsider the rating of our paper.

---

### Author Response · Authors · 2024-11-20
**To All Reviewers**

We sincerely appreciate the time and effort you have dedicated to reviewing our paper! Your valuable feedbacks have been carefully considered, and we will provide point-to-point responses to your reviews in respective reply. We remain open to any additional feedback you may have. If you feel it is appropriate based on our responses, we would be extremely grateful if you could consider raising our score.

---

### Author Response · Authors · 2024-12-04
**Concluding Response**

Dear Reviewers, AC, SAC, and PC,

We would like to begin by expressing our sincere gratitude for your engagement throughout the rebuttal process, which has significantly enhanced the quality of our paper.

Our work introduces two theoretically guaranteed practical algorithms, AFI-TM and AFI-EA, and a theoretical theorem AFI-OT. These are all based on the analytical formulation of the diffusion Fisher information. These algorithms enable us to perform NLL evaluation and adjoint guidance in a much more computationally efficient manner. These tasks were computationally very hard using naive methods.

In response to the feedback from reviewers, we have made the following revisions to our manuscript:
- **Clarification on proofs:** We have added detailed explanations for our derivations [QLo1].
- **Discussion on Hutchinson's trace estimator:** We have added a comparison and discussion on Hutchinson's trace estimator [QLo1][jSGi].
- **Discussion on related works:** We have added more discussion on related works that connect to our analytical formulation of Fisher in DMs [r2vX][QLo1].
- **Connection of AFI-OT to AFI:** We have added more discussion on the connection of our Theorem 1 to other parts of our paper [r2vX].
- **Syntax errors and typos:** We have revised typos and syntax errors as kindly pointed out [jSGi][r2vX].
- **Tighter bounds:** We derive tighter error bounds in Proposition 7 [jSGi].
- **Intuition of AFI-EA:** We add explanations on the intuition of our AFI-EA method [jSGi].
- **Experimental details:** We have added experimental details on the training costs and hyperparameters as kindly pointed out [iHvF].

We are immensely grateful for your feedback and suggestions for improving our manuscript. We kindly request your generous consideration of these points in the final evaluation of our paper.

Sincerely,

Authors of Submission 383

---

### Meta-Review · Area_Chair_qwxE · 2024-12-19

**Metareview:**

**Summary of Discussion:**
The paper addresses the analytical formulation of Fisher Information (FI) in diffusion models and proposes algorithms for likelihood evaluation and adjoint guidance sampling. While the reviewers appreciated the potential of these contributions, key concerns remained.

**Key Concerns:**

1. **Limited Theoretical Novelty:**
   - Proposition 3 and related theoretical results were found to be extensions or adaptations of prior work, limiting their impact.
   - Theorem 1, while corrected during rebuttal, offered minimal practical significance.

2. **Experimental Scope:**
   - Experiments lacked breadth and did not convincingly demonstrate the utility of the proposed algorithms across diverse scenarios.
   - Insufficient initial detail on training costs and hyperparameter tuning raised concerns about reproducibility.

3. **Presentation:**
   - Clarity and organization issues, particularly around linking theory to algorithms, reduced the paper's accessibility.
   - Language improvements during rebuttal were noted but were not sufficient to fully address concerns.

**Conclusion:**
While the paper tackles a relevant problem and proposes promising algorithms, the limited novelty, narrow experimental scope, and clarity issues ultimately do not meet the bar for acceptance. With expanded experiments and a more focused presentation, the work could contribute to future advancements in this area.

**Additional Comments On Reviewer Discussion:**

See above.

---

### Decision · Program_Chairs · 2025-01-22

Reject